# The Tharsis mantle source of depleted shergottites revealed by 90 million impact craters

A. Lagain [1✉], G. K. Benedix [1,2,3], K. Servis [1,4], D. Baratoux[5,6], L. S. Doucet [7], A. Rajšic[1],
H. A. R. Devillepoix[1], P. A. Bland[1], M. C. Towner[1], E. K. Sansom[1] & K. Miljković [1]

The only martian rock samples on Earth are meteorites ejected from the surface of Mars by asteroid impacts. The locations and geological contexts of the launch sites are currently unknown. Determining the impact locations is essential to unravel the relations between the evolution of the martian interior and its surface. Here we adapt a Crater Detection Algorithm that compile a database of 90 million impact craters, allowing to determine the potential launch position of these meteorites through the observation of secondary crater fields. We show that Tooting and 09-000015 craters, both located in the Tharsis volcanic province, are the most likely source of the depleted shergottites ejected 1.1 million year ago. This implies that a major thermal anomaly deeply rooted in the mantle under Tharsis was active over most of the geological history of the planet, and has sampled a depleted mantle, that has retained until recently geochemical signatures of Mars' early history.

[1] Space Science and Technology Centre, School of Earth and Planetary Science, Curtin University, Perth, WA, Australia. [2] Department of Earth and Planetary Sciences, Western Australian Museum, Perth, WA, Australia. [3] Planetary Sciences Institute, Tucson, AZ, USA. [4] CSIRO—Pawsey Supercomputing Centre, Kensington, WA, Australia. [5] Géosciences Environnement Toulouse, University of Toulouse, CNRS & IRD, 14, Avenue Edouard Belin, 31 400 Toulouse, France. [6] University Félix Houphouët-Boigny, UFR Sciences de la Terre et des Ressources Minières, Abidjan-Cocody, Côte d'Ivoire. [7] Earth Dynamics Research Group, TIGeR, School of Earth and Planetary Sciences, Curtin University, Perth, WA, Australia. ✉email: anthony.lagain@gmail.com

Martian meteorites are the only samples from the Red Planet available for laboratory analyses. More than 307 pieces of 166 unique samples, originating from at least 11 source craters, are curated in the world's collections[1]. Ejection ages, based on cosmic ray exposure (CRE), vary from 0.7 to 20 Myr[2–4]. The ejection sites are still unknown, despite several previous propositions[5–7], motivated by the significance of establishing a link between the crystallization ages, and the chemical and mineralogical properties of these samples with surface geology. Martian meteorites comprise five broad petrological categories: shergottites, nakhlites, chassignites, ALH 84001 and NWA 7034 and related pairs[4]. The shergottites are the most represented in terms of mass and number of specimens in collections[4]. Based on the concentration of rare earth elements (REE) and isotopic compositions, this group is subdivided into three distinct geochemical classes: depleted, intermediate and enriched[4]. REE patterns are inherited from the mantle source from which the rocks crystallized and subsequent magmatic processes such as fractional crystallization.

The ejection ages of depleted shergottites cluster around a value of $1.1 \pm 0.2$ Myr[2–4]. This suggests that they were ejected as sub-meter rock boulders by a single meteoritic impact. Although it would be possible that a rock sampled at a given place by a meteoritic impact had been previously transported by other processes (such as a previous impact with no evidence of shock), we will assume here that the rocks were generated at the impact location by magmatic processes. This group includes basaltic and poikilitic textures but is largely dominated by olivine-phyric shergottites and is characterized by a common depletion in REE. Crystallization ages range from ~330 Myr to ~570 Myr for 11 specimens, whereas NWA 7635 is ~2.4 Gyr old[4]. The temporary controversy regarding the significance of much older Uranium–Plomb ages for some of these samples is now being settled[8]. The hypothesis according to which the Mojave impact crater, located in old Noachian terrains, might be the source of all shergottites[5] can now be discarded. The variability of crystallization ages and peak shock pressures reported for this group of depleted shergottites launched 1.1 Myr ago[4] imply that a wide diversity of volcanic rocks was ejected from different geological units exposed at the surface of Mars or at shallow depths.

The formation of an impact crater generates debris ejected with speeds above and below the escape velocity on Mars (5 km/s). The fraction of ejecta material with a velocity higher than the escape velocity on Mars may get through the Martian atmosphere and into the interplanetary space. Numerical simulations suggest that impact events capable of producing such fragments would form craters larger than ~3 km in diameter on the Martian surface[9,10]. A portion of the material ejected with a velocity lower than 5 km/s (accounting for atmospheric deceleration) falls back to the surface in a radial pattern or rays around the primary source crater and forms secondary craters. Secondary craters reach a maximum diameter of about 2–5% of the primary crater diameter[11–13]. For instance, a 30 km crater would typically form secondaries smaller than 1 km diameter. These secondaries are shallower than those formed by primary impacts and are rapidly eroded. Considering an average depth/diameter for these craters of ~0.11[6] and an Amazonian (<3 Ga) crater obliteration rate of 100 nm/yr[14], a secondary crater of 100 m in diameter would be erased in 50 Myr (erosion of half of the depth and infilling of the other half). Therefore, the occurrence of radial patterns of small secondaries associated with a primary crater is a diagnostic feature of a recent impact[6,15,16]. Thus, the crater source of depleted shergottites launched 1.1 Myr ago should be associated with abundant small secondaries, in the ~10–300 m-size range. Identification of rayed craters is possible using images in the thermal infrared domain[5,16], but this approach is hampered by image resolution (100 m/pixel) and dust coverage (about half the Martian surface would remain not accessible by this technique). Existing databases of impact craters on Mars do not cover the range of diameters (less than 1 km) that are relevant to find radial patterns of secondaries associated with primary craters <30 km[11,12]. The use of high-resolution imagery would address this issue, but manual mapping of the tens of millions of secondary impact craters constellating the surface of Mars is not feasible.

In this work we adapt a Crater Detection Algorithm (CDA)[17] to detect craters <1 km on the whole surface of Mars. We build a database of 90 million impact craters and identify secondary crater rays system to locate the crater candidates responsible of the ejection of martian meteorites. We show that Tooting and 09-000015 craters are the most likely source of the depleted shergottites ejected 1.1 Ma ago. We discuss the relationship between this group of meteorites and the Tharsis volcanic province, where these two craters are located. We infer the presence of a major thermal anomaly, deeply rooted in the mantle, under the Tharsis dome. This thermal anomaly has sampled a depleted mantle over most of the geological history of Mars.

## Results

**Machine learning approach to pinpoint the meteorite crater sources.** We retrained a Convolutionnal Neural Network[17] (Methods, subsection The Crater Detection Algorithm) to identify craters down to 25 m in diameter across the entire surface of Mars. The algorithm was trained using High-Resolution Imaging Science Experiment (HiRISE) images (25 cm/pixel) and applied on the global Context Camera (CTX) mosaic[18] (5 m/pixel), thus generating a database of ~90 million detections (Methods, subsection Application to the CTX global mosaic and evaluation, Supplementary Table 1 and Supplementary Figs. 1–4). To visualize secondaries, and therefore recent primary of different sizes, the density of secondaries for three ranges of sizes ($150 < D < 300$ m, $75 < D < 150$ m and <75 m) are represented as a map (Figs. 1 and 2) using, respectively, the red, blue and green channels (Methods, subsection Crater Density map computation, and Supplementary Fig. 5). The distribution of detections >300 m are not examined as the presence of large secondaries is not a discriminant factor for the (young) age of the primary crater. A careful survey of this map allowed the identification of 19 secondary ray systems associated with large and recent primary craters (Fig. 1).

**The ejection site of the depleted shergottites.** The ages of these 19 young impacts are determined using manual crater counts (excluding secondaries) on their ejecta blanket (Methods, subsection Model age derivation and uncertainties) and are used to test the completeness of the survey presented here: these 19 impact craters may represent the complete record of large and recent impact craters on the surface of Mars. Despite the inherent lack of small craters on several ejecta blanket that might bias the formation model age of those craters (Methods, subsection Model age derivation and uncertainties), we found that the 17 craters larger than 7 km in diameter and younger than 10 Myr old plot on an $8.2 \pm 2$ Myr isochron (Supplementary Fig. 6 and Supplementary Table 2). There is only 9% of chance that a crater larger than 7 km was missed according to Poisson uncertainties[19,20] and primary crater production function[21]. This suggests that the use of secondaries as a criteria to identify recent craters is valid and that the identified crater population larger than 7 km formed in the last ~10 Myr is complete. Thus, the visualization of 90 million craters counted using the automatic detection approach[17] allows us to reduce the number of potential source craters of the Martian

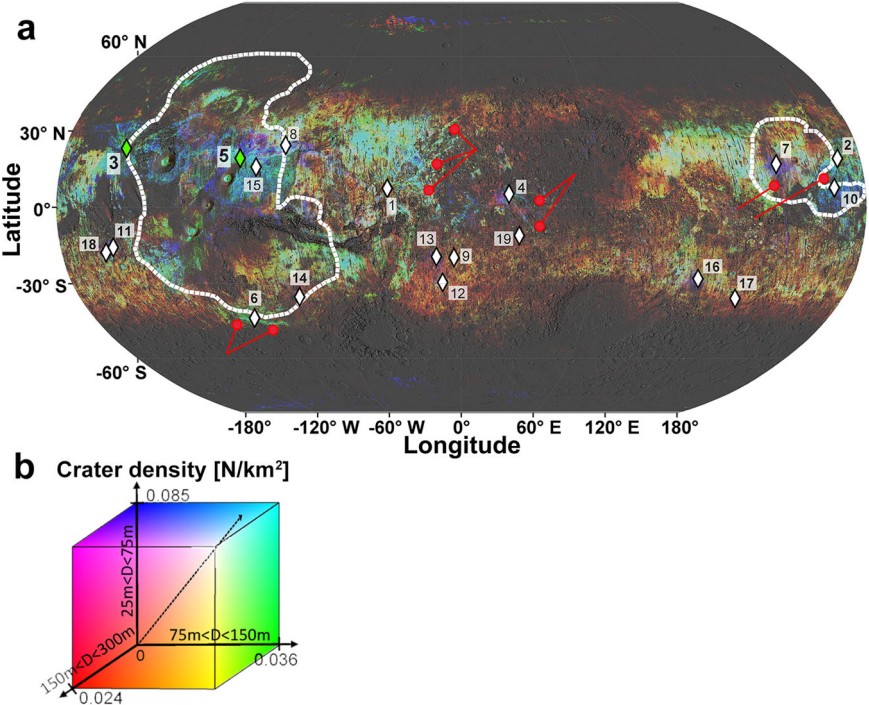

**Fig. 1 Crater density map of Mars. a** Density map of craters <300 m in diameter (89,054,458 entries), resolution 0.05°/px. Colours indicate crater densities of specific diameter ranges (alternatively, supplementary Fig. 5 presents the crater density for each band separately). The diamonds identify 19 potential crater candidates (D > 3 km) for the launch of Martian meteorites (see Supplementary Table 2 for the size, location and model age of those craters), identified from radial patterns of secondary crater rays (D < 300 m). White dashed lines represent the contour of the Tharsis dome (at the left) and Elysium (at the right). Tooting (label 3) and 09-00015 (label 5) are highlighted in green and their rays of secondaries are shown on Fig.2a and b respectively. Red arrows point some secondary crater rays and readers are invited to visit http://craters.computation.org.au/ and http://HIVE.curtin. edu.au/research/CDA-94M-release for high-resolution versions of the map. Background: MOLA shaded relief (http://bit.ly/HRSC_MOLA_Blend_v0), projection: Robinson. **b** Legend of the crater density map shown on panel a. Each dimension of the colorcube corresponds to the crater density (number of craters per km$^2$) of specific diameter ranges (blue: 25–75 m, green: 75–150 m and red: 150–300 m).

meteorites from ~80,000 (the number of craters >3 km on the surface[9–12]) to 19 with 91% confidence.

To further restrict the number of candidate crater sources for the depleted shergottites launched 1.1 Myr ago, their crystallization ages (~330 Myr–~570 Myr and ~2.4 Gyr[3,4]) are compared to the ages of terrains surrounding each crater. This approach assumes that crater counts on the area surrounding the crater source may provide the age of the volcanic episode comparable to the crystallization age of the meteorite. Surrounding terrain model ages have been derived for all impact craters using the recently revised Mars crater catalogue[12], compiled manually and complete down to 1 km[11,12], completed for higher precision by manual mapping of smaller craters if necessary (Methods, subsection Model age derivation and uncertainties). Model ages obtained for different impact cratering rates, ranging between a factor of 2 around the reference impact flux over the last ~3 Gyr[21] were also estimated (Methods, subsection Influence of impact cratering rate uncertainties on model ages, Fig. 3, and Supplementary Table 2).

Among the 19 craters with small secondaries, only two craters, Tooting (Fig. 2a) and 09-00015 (Fig. 2b), have model ages compatible with the range of crystallization ages of the depleted shergottites (Fig. 3). They are both located on volcanic terrains interpreted as stacking of lava flows with thicknesses of up to a few kilometres in some places[22] and associated with Tharsis' dome activity, the largest volcanic province on Mars. We note that Tooting crater is the only one that matches both ejection and crystallization ages. Based on our crater counts on Tooting's ejecta blanket, its formation model age is estimated to be ~1 Myr, while our counts on the surrounding ground gives a model age of

308 ± 41 Myr. Moreover, accounting for large craters around Tooting that have been filled-up by recent volcanic activity, an Early Amazonian model age can be derived ($1.77^{+0.69}_{-0.58}$ Gyr). Although not discriminant, this age matches within error the crystallization age of NWA 7635[3] (Fig. 3 and Supplementary Figs. 7 and 29) and might represent a minimum age of older volcanic episodes that have been subsequently covered by younger Amazonian lavas.

Tooting crater (D ≈ 30 km) is the second largest of the 19 rayed craters younger than 10 Myr old. Impact simulations of the Tooting crater formation (Methods, subsection Impact cratering simulation, and Supplementary Fig. 46) predict the excavation of a volume of ~1.28 × 10$^{12}$ m$^3$ (Supplementary Fig. 47), assuming excavation was made by a vertical impact direction. Similarly, the impact simulations were made for five other crater candidates. Supplementary Fig. 46 shows the moment when the transient crater was reached in each case. The transient crater rim volume was used as an estimate the volume of excavated materials, summarized in Supplementary Fig. 47. Here we used the total excavated volume to quantify the increasing of the amount of ejected material with the crater diameter. This is relevant assuming a constant fraction collapses back into the crater and a fraction is ejected out of the crater, potentially forming secondary craters. Compared to the two other craters with young surface ages, (~3.97 × 10$^{11}$ m$^3$ and ~6.67 × 10$^{10}$ m$^3$ for 09-000015 (D ≈ 20 km) and Zunil (D ≈ 10 km) respectively), Tooting has set in motion ~3.2 and ~19.3 times more material. Tooting produced a relatively larger volume of material responsible for secondary cratering and rocks escaping Mars' gravity compared to 09-000015 and Zunil. Further, the ejecta blanket of Tooting exhibits

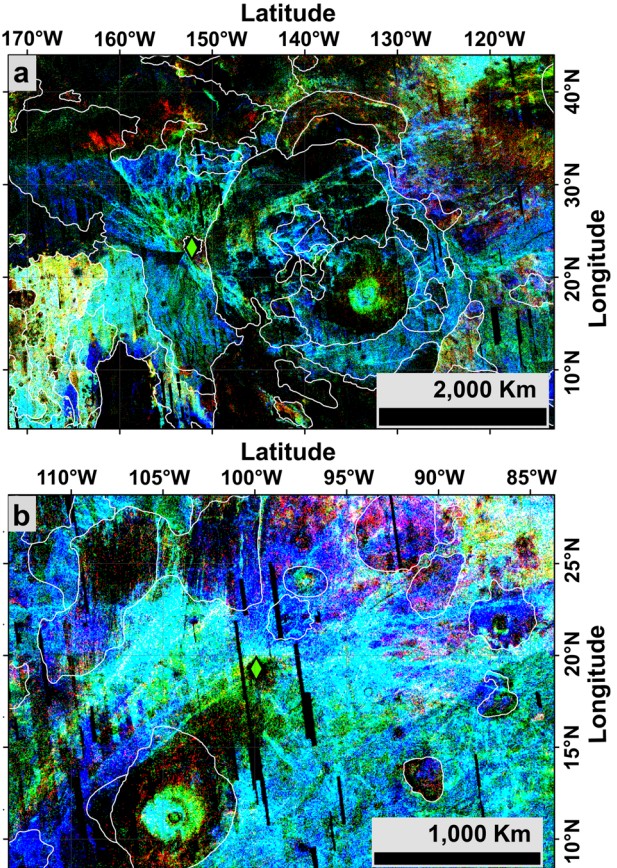

**Fig. 2 Close-up of cratering density around the two most likely crater source of the depleted shergottites. a** Tooting and (**b**) 09-00015 craters (respectively label 3 and 5 on Fig. 1a). The location of the two craters are shown as green diamond. White outlines are contacts between geological units[44]. The readers are referred to Fig. 1b for the legend of the cratering density.

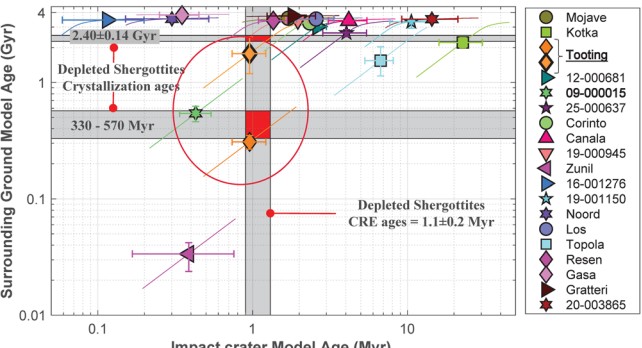

**Fig. 3 Model ages of the crater candidates ages and their host units.** Vertical and horizontal grey bands correspond respectively to the range of CRE ages[3,4] and the crystallization ages[3,4] of depleted shergottites launched 1.1 Myr ago including NWA 7635 (2.40 ± 0.14 Gyr). Error bars are generated based on crater counts. Oblique colour lines illustrate the expected displacement in time if different cratering rates are considered (range of a factor of 2 around the reference impact flux[21]). The red circle outlines the two craters whose model age of their host unit is compatible with the range of crystallization age of depleted shergottites launched 1.1 Myr ago: Tooting and 09-000015. Note that two ages are reported for Tooting, one (308 ± 41 Myr) corresponding to immediate surrounding (thin outline), and the other ($1.77^{+0.69}_{-0.58}$ Gyr) accounting for large craters around Tooting and partially filled-up by more recent lavas (thick outline). Craters in the legend are sorted by size (Supplementary Table 2).

For all these reasons, Tooting crater appears to be the most likely source of the depleted shergottites, even though we cannot rule out 09-000015 crater. It is important to note that both craters are located on the Tharsis dome. Therefore, even if two candidates remain, it is possible to link the source of depleted shergottites to the Tharsis dome's magmatic activity. We establish a link between Tharsis and rock samples, which opens the possibility to discuss geodynamic context and mantle source of the largest volcanic edifice in the Solar System, based on geochemical constraints, including isotopic data.

## Discussion

Long-lived ($^{147}$Sm-$^{143}$Nd, $^{87}$Rb-$^{87}$Sr, $^{176}$Lu-$^{177}$Hf, $^{187}$Re-$^{188}$Os, $^{233,238}$U-$^{232}$Th-$^{206,207,208}$Pb) and short-lived ($^{146}$Sm-$^{142}$Nd, $^{182}$Hf-$^{182}$W) isotopic composition of the depleted shergottites[4] indicate, respectively, that they sampled a highly depleted mantle formed early in Mars history, during the differentiation of the Martian magma ocean (MMO), ~4.5 Gyr ago[4,27]. In addition, considering that olivine phenocrysts in the ol-phyric depleted shergottites are in near-equilibrium with their parental melt, the potential mantle temperature (Tp) can be estimated from 1714 to 1835 °C[28,29]. This is hotter than estimates from in situ rock analyses in Gusev (1300°–1500 °C), Meridiani (1400 °C) or Gale crater (1250–1500 °C)[28–30]. On Earth, the broad range of potential mantle temperatures reflects the diversity of mantle melting environment, such as hot spots or large igneous provinces above mantle plume, mid-oceanic ridges or fluid-enhanced melting at subduction zones. The high Tp inferred from ol-phyric depleted shergottites defines an adiabatic gradient that crosses the Martian fertile mantle (primary) solidus[28,31] at depth >1000 km (Fig. 4a). This indicates that melting potentially started in the transition zone, marked by the appearance of γ-spinel (ring-woodite) or below in the lower mantle. However, direct comparison between those Tp and a fertile mantle solidus, with Mg# of 0.75–0.77, can be tenuous as the latter would melt at lower temperature[32], with respect to a depleted mantle source characterized by higher Mg# of 0.85–0.86[33]. Nevertheless,

fluidized morphologies (as well as 09-000015), indicating the presence of subsurface volatiles that increase the spallation volumes by as much as 10%[9]. This crater also presents an asymmetric distribution of secondary craters rays, a feature not seen in 09-000015 and its rays system extends at least up to 1500 km from the crater, i.e. up to 100 crater radii (Fig. 2a). This is more than 1000 km farther than previously noted[7]. The forbidden zone (zone lacking secondaries) observed on Fig. 2a confirms an oblique impact[23]. Based on the morphological evidence of the crater[7,23] and numerical simulations[24], the angle of entry of the impactor that formed Tooting crater ranged between 30° and 50° from the surface. Numerical models[24] suggest that such an oblique impact would enhance the fraction of ejected debris reaching escape velocity, all other parameters being equal. For instance, the ejected mass from a 45° impact is more than 7 times greater than for a vertical impact[24].

Regarding the surface environment, analysis of volatiles in impact melt pockets of the Tissint meteorite shows post-magmatic and pre-impact low alteration by subsurface water, which equilibrated with the present-day atmosphere[25]. Both the lobate morphology of the ejecta of Tooting and the flow textures in its inner wall[26], interpreted to be sediments remobilized by water seeping, argue for the existence of subsurface reservoirs of volatiles at the time of impact. Microscopic evidence for recent subsurface reservoirs of water on Mars can thus be linked to surface morphology.

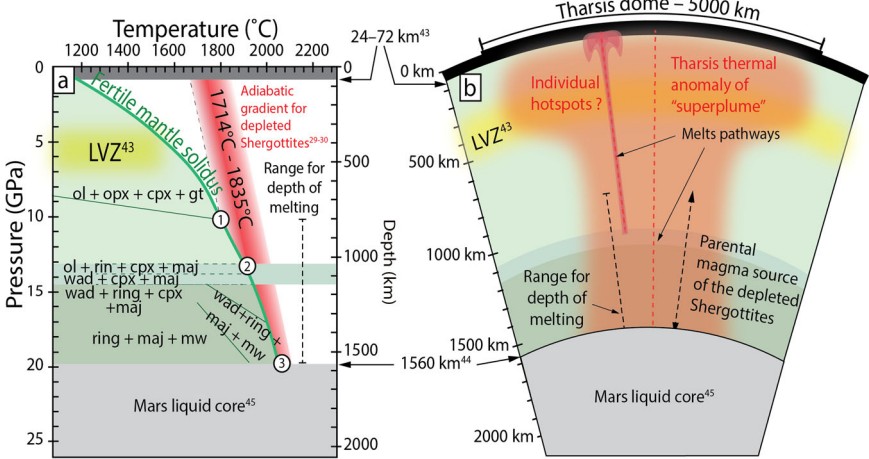

**Fig. 4 Melting environment for the formation of the depleted shergottites. a** Pressure-temperature-depth phase diagram for the Martian mantle[28]. The adiabatic range for the depleted shergottites (in red) extrapolated using mantle potential temperature[28,29,32], suggests that melting potentially occurs below 10 GPa (1), within the transition zone (2) and possibly down to the core-mantle boundary (3), i.e. between ~800 and ~1600 km. **b** Schematic cross-section of Mars below the Tharsis dome, showing the Tharsis superplume beneath the crust as well as the depth range of melting for the parental magmas of the depleted shergottites and potential melts pathways (red dashed lines). Structure of the Martian interior and layers thicknesses are inferred from SEIS (Seismic Experiment for Interior Structure) data on board of the InSight lander[80–82]. ol olivine, opx orthopyroxene, cpx clinopyroxene, gt garnet, rin ringwoodite, wad wadsleyite, maj majorite, mw magnesio-wustite.

experimental results show that the composition of melt extracted from a fertile mantle at 8 and 10 GPa match the high magnesium content of some depleted shergottites[33], which is consistent with melting starting at least 10 GPa, i.e. at the bottom of the upper mantle (Methods, subsection Potential mantle temperature and depth of melting).

These data support a hot-spot origin for the formation of the parental magma of these meteorites[34–39], with a partially molten hot section of the mantle formed relatively early in the history of Mars. The most accepted explanation for the origin of the Tharsis volcanic province is the superplume hypothesis[34], with the onset of a large thermal anomaly deeply rooted in the mantle since the multi-stage crystallization of the MMO, ~20 Myr after the accretion[35,36]. In absence of plate tectonics, this abnormally hot mantle would produce shield volcanoes fed by a single or multiple plumes[34] (Fig. 4b) that have builded-up Tharsis at least over the last ~4 Gyr[37–39]. The Early Noachian stage of the Tharsis growth is thought to have been intense enough ($50 \times 10^6 - 100 \times 10^6$ km³ within 500 Myr[39]) to lead to a True Polar Wander, i.e. a rotation of the crust in respect of its spin-axis responsible for its present equatorial position[34,40]. The link between the depleted shergottites and the Tharsis dome suggests that Amazonian volcanic activity (since 2.4 Gyr until 330 Myr ago, at a minimum) related to the Tharsis bulge formed above the superplume are sourced deep in the mantle (Fig. 4b) since that time, and by extrapolation, perhaps since the beginning of the Tharsis formation.

Taking into account the size of Tharsis volcanic province (up to 25% of the total surface of the planet), our results suggest that a large portion of the Martian mantle is highly depleted, relatively possibly dry[41], relic of Mars' early history and contributes to a better understanding in the geochemical and geodynamic structure evolution of Mars. Such Amazonian activity reflects that some portions of Mars' interior are anomalously hot, which might induce anomalously slow velocity waves. This hypothesis may be tested through ongoing measurements performed by the InSight mission.

## Methods

**The Crater Detection Algorithm.** The Crater Detection Algorithm has been previously described in detail[17]. The key features are summarized here followed by a description of the adaptation achieved for the purpose of this study.

The CDA was initially trained on Thermal Emission Imagery System (THEMIS) Day IR images (100 m/px) covering 1762 impact craters ≥1 km in diameter from the Mars crater database[12]. To detect smaller impact craters and thus be able to identify rays of sub-kilometre-sized secondary craters, higher resolution datasets, such as HiRISE (High-Resolution Imaging Science Experiment) and CTX (Context Camera) are necessary. The high degree of detail found in these datasets makes the current training for the CDA inefficient due to the presence of decameter to hectometer-sized circular structures that are not of impact origin. This include landscapes within field dunes exhibiting circular features or structures formed by the erosion, essential to include in the training dataset to avoid detecting them. Retraining the algorithm is therefore essential to accurately detect impact craters smaller than 1 km in diameter.

Two significant hurdles were required for retraining: (1) creating a new set of labelled images, i.e. a training library where all impact craters visible on a set of images are identified using HiRISE imagery[42] and (2) increasing the speed of the training-validation and test steps.

The first challenge was achieved using the HiRISE mosaic[42] of the Jezero crater (E77-5-N18-0) where 2142 craters have been manually identified in the aim to train the CNN (Convolutional Neural Network), of which 550 have been held out for validation. This labelled dataset was also augmented by applying a range of transformations (rotate, shear, scale and translate) using YOLOv3[43].

To address the second challenge, we updated the CDA to achieve a higher modularity of the components, but at the same time made the software more user friendly and able to run both on a desktop for development as well as the High Performance Computing (HPC) clusters at the Pawsey Supercomputer Centre in Western Australia. This is particularly critical regarding the test step as well as the scoring of the entire CTX mosaic, when a few hundred gigabytes of imagery datasets (>5 TB in the case of the CTX global mosaic[18]) must be analysed in a timely fashion. To that end, we broke down all the processing steps (image reprojection and downsampling, tiling, scoring) of the previous version of the CDA[17] to individual tasks, all with their own container. Thus, each of the tasks that can be run individually and in parallel for different images. Each of which goes through all the individual steps producing crater locations in geographical coordinates of the original GeoTIFF image (using the spatial reference system specified therein). Before the detection, each HiRISE image is divided into tiles (960 × 960 pixels) and analysed three times by the CNN at native and downsampled resolution (native: 0.25 m/px, downsampled: 2 m and 10 m/px). This ensures that larger craters that might be bigger than a single tile at native resolution are detected[17]. When all the individual images have been processed, a non-maximum suppression (NMS) algorithm is performed on the union of the results to remove potential double-counted craters and to obtain the final data product. Non-maximum suppression consists of identifying duplicate detections and rejecting all but the one with the highest confidence score. In this instance we consider duplicates to be detections that overlap with an intersection over union ratio of more than 0.3.

**Application to the CTX global mosaic and evaluation.** The Context Camera on board of the Mars Reconnaissance Orbiter (MRO) allowed to build the highest global mosaic currently available[18]. Its resolution of ~5 m/px allows us to identify

impact craters as small as 25 m. The mosaic covers about 99% of the surface of Mars between 88°S and 88°N of latitudes. Each of the 15840 tiles composing the mosaic covers an area of 4° × 4° and has been reprojected using gdalwarp from equirectangular to stereographic projection to avoid geometric distortion, which is especially pronounced at high latitudes. Each image is then downsampled in resolution from the raw resolution (5 m/px) to 40 and 160 m/px. The image is stored in GeoTIFF format. This step takes about 66 CPU hours but running in parallel reduces the computation time to generate the 15840 images that compose the entire CTX mosaic in 45 min. This set of images has been finally analysed by our retrained CNN in about 24 h using 12 nodes from Pawsey Topaz cluster featuring dual Tesla V100 GPUs and another 3 h for the NMS phase using one node from the Pawsey Magnus cluster. This results in 94,773,450 crater detections, of which just 89,054,458 are smaller than 300 m in diameter and used to analyse the spatial crater distribution presented on Fig. 1a.

To statistically validate the results, we manually mapped ~2000 craters using CTX imagery on different types of terrains and compared them to the CDA's results. Supplementary Table 1 shows how many craters have been manually identified for each terrain as well as the number of true positive (TP), false detection or false positive (FP) and missed craters or false negative (FN) for different diameter ranges from D > 60 m. We cannot guarantee the validity of the manually mapped crater population <60 m and the associated metrics due to the resolution of the image. We calculated the true positive rate (or recall) defined as $\frac{TP}{P}$, the precision $\frac{TP}{TP+FP}$ and F1 score $\frac{TP}{TP+\frac{1}{2}(FP+FN)}$. The recall expresses the ability of the CDA to find all relevant craters in our dataset, while the precision expresses the proportion of relevant craters detected by the CDA. F1 score combines both metrics using the harmonic mean of the recall and precision, thus allowing to punish extreme values. From this, the average precision, recall and F1 are higher than 0.75 for craters larger than 70 m in diameter except on highlands and at high latitudes (>50°) where those metrics are lower than 0.75 but always above 0.6. Detections at mid and high latitudes (>50°) are of the lowest quality compared to those closer to the equator, mostly due to the degraded morphology of impact craters. We note that merging the metrics between each type of terrains (last column of Supplementary Table. 1) is not representative of the overall performance of our CDA on the whole surface of Mars since it is not normalized by the fraction of the surface represented by each type of terrains. For example, aprons, impact or high latitudes constitute a minor fraction of the surface[44]. Also the total recall and F1, accounting for craters <60 m (Supplementary Table 1) are an estimation if the manually mapped craters are complete down to the smallest crater detected by the CDA. This is certainly not the case, mostly due to the resolution of the image and those values cannot be used for evaluating the performance of the algorithm.

We also evaluated the precision of the diameter estimation among true positive detections (913 across all terrains considered in this evaluation test). Supplementary Figure 2 shows the percentage difference between the diameter estimated by the CDA ($D_{TP}$) and the diameter manually measured here called ground truth ($D_{GT}$). We observe an increase in the overestimation with the decrease in crater size due to the resolution of the image. This is especially true for craters <50 m. For craters larger than 100 m in diameter and except rare instances at high latitudes, the CDA estimates crater size with errors as much as 25%. Supplementary Figure 3 presents a kernel density plot of the Supplementary Fig. 2, where three close-ups of impact craters detected by the CDA illustrate different precision in the diameter estimation.

Our validation test shows that our database is statistically robust and complete from 100 m in diameter. The CDA estimates crater diameter with a precision of ~25% for craters larger than 100 m. While the counting accuracy decreases with decreasing diameter (in particular close to 10x the native resolution of the image, i.e. 50 m), true positive rates are equal to or better than average manual counting differences between two humans[45] (~±30%) for craters larger than 70 m in diameter (Supplementary Table 1). Degraded crater morphologies, mostly represented at mid and high latitudes (>45 degrees), are the main source of error in absolute counts and diameter estimation compared to those closer to the equator, as described in the work presented in a previous version of our algorithm[17]. Also, while the number of detections in the 25–75 m is 32,889,699 (corresponding to one third of the total detections), the number of craters detected within the higher bracket of this range (60–75 m, $N = 16,615,952$) is approximately equal to those smaller than 60 m ($N = 16,273,747$) (see Supplementary Fig. 4 for a plot of the Crater Size-Frequency Distribution of the detections). The population of small craters <60 m, where the diameter is generally overestimated up to 50% does not dominate the population of craters used to populate the blue band of the density map shown on Fig. 1a.

We note that our catalogue accounts for nearly all craters down to 70–100 m diameter range as shown on Supplementary Fig. 4 and fills the gap of sub-kilometre craters existing in manual crater database[11,12]. Nevertheless, since craters >200 m constitutes a minor portion of the detections tested in the present work, an extensive evaluation of larger craters is needed and left for future investigation. It is of note that the usual caveats and limitations of an automatic algorithm, including proportion of false positives, have negligible consequences on recognizing the main systems of secondary craters, which is the purpose of this study. Given the large numbers of secondaries associated with a given primary, a complete dataset is not required when using the CDA as a visualization tool, and the performance of the CDA is appropriate for the detection of secondary crater field.

**Crater density map computation**. The crater density map presented on Fig. 1a has been computed using Matlab by defining a grid where each cell is 0.05° × 0.05°. The number of craters contained in each cell and falling into each of the diameter range (25–75 m, 75–150 m and 150–300 m) is computed. These values are then divided by the area of each cell using the mars ellipsoid to obtain the crater density for each cell (Supplementary Fig. 5). The division of the data in different ranges is achieved to make sure to identify primary craters to different sizes. Results are stored in a georeferenced RGB raster and loaded on ArcGIS (ESRI) to adjust the stretch of the values in each histogram band. For each band, values between the average ±2 standard deviations are stretched between 0 and 255 (values beyond this range are assigned either 0 and 255). The final product is the RGB crater density map presented on Fig. 1a (see Supplementary Figs. 8–26 for a close-up of the crater density around each crater candidate), and available at high resolution on a CESIUM interface at http://craters.computation.org.au/, accessible upon registration, and at http://HIVE.curtin.edu.au/research/CDA-94M-release.

**Model age derivation and uncertainties**. We derived the formation model age of the 19 impact craters exhibiting rays of small secondaries (<300 m) by counting craters superposed on their ejecta blankets and excluded areas corresponding to the impact crater floor and its rims[46] using the CraterTools software[47]. In order to determine the age of these young craters, we need an accurate crater count superposed on the ejecta of these craters[48]. Given the young age of these surfaces, a small number of craters are expected, and the result is very sensitive to any mis-identification of craters (missing craters or false detections). Therefore, CDA results were used as a first pass to pinpoint the primary impact craters superposed on each ejecta blanket and were then corrected and completed by a manual verification count. The age of terrains surrounding each large crater has been derived using the manual crater database complete down to 1 km in diameter[11,12] where secondary craters are marked and discarded from the counting.

Each Crater Size-Frequency Distribution was loaded into CraterStats II[19] and fitted with an isochron using a standard chronology model[21]. We derived all model ages using the differential representation[49,50] defined at a given diameter Dx as $N(Dx) = n(Dx1, Dx2)/(Dx2 − Dx1)$. Compared to cumulative plot, this method allows to plot each point independently from subsequent diameter bins, and an easier recognition of a resurfacing events contribution or a contamination by secondary craters[49]. However, the binning of the data can biased the model age derivation, when an isochrone is fitted to the CSFD[20], in particular when few impact craters are used. We used the Poisson timing analysis technique[20], allowing an exact prediction of the crater chronology model, whatever the chosen binning technique.

Model ages derived from crater counts can be biased due to the variability of the impacted material physical properties[51–53]. Two asteroids of the same size and approach conditions (velocity and angle) will produce an impact crater with two distinct final diameters depending on whether the impacted material is made of consolidated or unconsolidated rocks. This variability may affect the entire CSFD. The target properties dependency in the final crater size is more important for craters <1 km in diameter. Significant physical properties variability between ejecta deposit, composed of brecciated and unconsolidated material, and the surrounding terrains are expected. However, the thickness of the ejecta blanket of a 10 km Layered Ejecta Rampart Sinuous crater[54] does not exceed a dozen meters[55], and is up to 20 m in the case of Tooting crater[23]. The dominant fraction of the excavated volume following the formation of small impact craters on those ejecta blanket, typically larger than a few hundred meters in diameter, is therefore represented by the underlying rocks. Primary impact craters size used to derive model ages shall be dominated by the physical properties of the underlying rocks, rather than that of the ejecta. We therefore neglect ejecta layer properties effects on the crater SFD used to derive the model ages we present in this study.

The statistical effect of small counting area and/or sparse number of craters used to derive model ages of a planetary surface has been widely discussed in the literature[51]. While crater counts performed on areas larger than 100 km² are less sensitive to statistical bias due to the stochastic nature of the cratering record, the accuracy of the model ages decrease with decreasing count area size. This source of uncertainty is partially attenuated thanks to the use of the Poisson timing analysis technique[20] that allows meaningful model age estimates, even in the case where a unit on Mars does not exhibit any craters (see Supplementary Figs. 27–45 presenting the crater count results for each crater candidate and their surrounding ground). The host terrain of Zunil crater has been determined by crater counts down to 200 m in diameter due to the lack of kilometric craters surrounding this crater (Supplementary Fig. 36). Our result is consistent with previous estimates[56].

**Influence of impact cratering rate uncertainties on model ages**. Uncertainties of model ages derived from crater counts have been widely discussed in the literature[49,51]. The impact cratering rate over the last 3 Gyr is assumed to be constant but is poorly constrained due to the lack of lunar samples younger than 3 Gyr old. Also, the $R_{bolide}$ (factor used to convert the Lunar chronology to Mars[57,58]), has been derived from direct observations of the crater population and dynamical considerations of asteroid and comets population[59–61]. One of the largest source of uncertainty is the estimation of the respective contribution of cometary materials to the cratering record on the Moon and on Mars[60]. Model age

error bars reported in many studies are only based on the number of craters counted on a specific region; they do not consider the intrinsic uncertainty of the impact cratering rate involved in the chronology systems used to derive them. To account for these potential sources of variability in the model ages we report in this study, we varied the impact rate by a factor of 2 above and below the reference rate[21] between 0 and ~3 Gyr and recomputed all model ages of impact craters and their host units. Results are presented on Fig. 3 in the form of curves (straight lines for young ages) intersecting each dot, where the top right end corresponds to the combination of model ages obtained for an impact cratering rate two times lower than that generally assumed[21], and the lower left end of the curves correspond to a cratering rate two times higher.

**Impact cratering simulation**. We used the shock physics code iSALE-2D (https://isale-code.github.io/) to model complex impact craters. The iSALE-2D code[62–64] is a multi-rheology, multi-material extension to the finite-difference hydrocode[61]. Six numerical models were made to simulate the complex crater formation with a diameter ranging from 4 to 28 km. The impactor diameter was varied between 0.5 and 4.5 km, and impact velocity was kept constant. In all simulations, the projectile and the target were modelled using the analytical equation of state for dunite and basalt, respectively[65]. The strength model for both target and projectile was for a typical damaged rock[62]. Simulations were run until transient crater was formed (Supplementary Fig. 46). Transient crater size was defined as the maximum volume of the excavation cavity[66,67]. The final crater size is calculated from the impact scaling laws[61,68]. In these simulations, the displaced target material was tracked by setting up Lagrangian tracer particles in each cell. These tracers recorded information about cell density at every timestep (see Supplementary Tables 3 and 4 for a complete list of impact parameters used in iSALE simulations). Volume of the displaced material was measured above surface level and it was defined as rim volume. Rim volumes were then compared to the size of the final crater diameter to obtain the rim volumes of Tooting, 09-000015 and Zunil craters (Supplementary Fig. 47). Our results are consistent with impact crater scaling relationships.

**Potential mantle temperature and depth of melting**. The most straightforward way to estimate the depth of melting for a magma produced at pressure higher than 3 GPa is to (1) estimate the potential mantle temperature[69] (Tp) and (2) project the adiabat on the solidus of the mantle on a Pressure-Temperature (P-T) phase diagram (Fig. 4a). Note that for pressure <3 GPa, pMELTS[70] can be used (allowing thermodynamic modeling of phase equilibria in magmatic systems), but it has been calibrated on the composition of the terrestrial mantle[70]. Tp may also be estimated based on Mg–Fe partitioning between olivine and mafic-ultramafic melt, which is temperature dependent[71,72]. There are several sources of uncertainty using these methods, including the composition of the mantle source, the composition of the parental magmas at the solidus or the effect of fractional crystallization. However, studies on Earth mafic samples appear to be consistent with experimental results and geophysical observation, and therefore provide reliable information on the geodynamic setting on the source region of the mafic rocks[71–74]. The challenge in estimating the Tp for mantle mafic rocks is the uncertainty on the composition of the Martian mantle itself, in particular for non-refractory major elements. However, over the last decades, various studies came up with average compositions that are consistent with geophysical interpretations and argue for an iron-rich mantle (Mg# = molar ratio: Mg/(Mg + Fe), near 0.75[75–78]). Recently, the solidus of such iron-rich Mars mantle has been extended down to the core-mantle boundary, up to 25 GPa[32] which allows us to investigate partial melting of the mantle from top to bottom (Fig. 4b). In this study, we collected the Tp for depleted shergottites (1714–1835 °C), but also the Gusev (1300–1500 °C), Meridiani (1400 °C) and Gale crater (1250–1500 °C) estimated by refs. [28–30], which were calculated, at first, assuming a melt in equilibrium with a mantle with a Mg# of 0.86. The projection of the adiabat for the depleted shergottites on the P-T diagram predict that melting could start as deep as >13 GPa (>1000 km). However, this P-T relationship has been calibrated assuming a Mg# of 0.75 for the Martian mantle[32]. The melting temperature of a more refractory mantle with Mg# of 0.86 would be higher than that of a mantle[32] with Mg# of 0.75. Nevertheless, experimental results show that the composition of melt extracted from a Mg# of 0.75 mantle at 8 and 10 GPa match the high magnesium content of olivine-phyric depleted shergottites[33,79], which is consistent with melting starting at least 10 GPa, down to the lower mantle of Mars.

## Data availability
Correspondence and material requests should be addressed to Anthony Lagain at anthony.lagain@gmail.com. Remote sensing data (HiRISE and CTX imagery) are available at http://murray-lab.caltech.edu/CTX/tiles/ and at http://murray-lab.caltech.edu/Mars2020/. The crater density map presented on Fig. 1a is available at high resolution at http://HIVE.curtin.edu.au/research/CDA-94M-release and at http://craters.computation.org.au/. The algorithm training dataset that support the findings of this study is available in Zenodo with the identifier: https://doi.org/10.5281/zenodo.5514313. Derived data supporting the findings of this study are available within the paper, the Supplementary Information file, in Supplementary Data, and from the corresponding author A.L. on request. Source data are provided with this paper.

## Code availability
The Crater Detection Algorithm code supporting the findings in this study is available upon request from the corresponding author A.L. The numerical impact crater formation were made using the iSALE shock physics hydrocode. At present, iSALE is not fully open source. Application for use of iSALE can be made via https://isale-code.github.io/. Any recent stable release can be used to reproduce the data presented. We used the IDL 5.2 software (L3Harris geospatial https://www.l3harrisgeospatial.com/Software-Technology/IDL) to run the CraterStats II software available at https://www.geo.fu-berlin.de/en/geol/fachrichtungen/planet/softwarealgorithm and the ESRI's ArcGIS 10.8.1 software suite (ESRI https://www.esri.com/en-us/arcgis/about-arcgis/overview) and Matlab (https://au.mathworks.com/products/matlab.html) to produce the maps.

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

## Acknowledgements

A.L. and G.K.B. are funded by the Australian Research Council (DP170102972 and DP210100336), the Australian and Western Australian Government, Curtin University, and supported by the Pawsey Supercomputing Centre and ADACS (Astronomy Data and Compute Services). K.S. is supported by the Pawsey Supercomputing Centre,

ADACS, and CSIRO. Raw data were generated at Pawsey Supercomputing Centre. We thank Wesley Lamont and the Curtin Hub for Immersive Visualization and eResearch (HIVE) for their help in the visualization of our crater detection having allowed the best use of our algorithm.

## Author contributions

A.L. and G.K.B. are responsible for study conceptualization. K.S. adapted and retrained the CDA under A.L. guidance. A.L. performed the CDA evaluation, crater counts, age measurements, impact flux simulation, density map computation and secondary rays survey. D.B., L.S.D. and A.L. interpreted the potential mantle temperature data. A.R. and K.M. performed iSALE impact crater modelling. H.A.R.D., E.S. and M.T. helped with data handling and some aspects of the crater density map computation. A.L., G.K.B., D.B., L.S.D. and P.B. were involved in data interpretation. A.L wrote the manuscript, with contributions, review and editing from all co-authors.

## Competing interests

The authors declare no competing interests.
