## [Peer Review File · Nature Communications]

REVIEWER COMMENTS

Reviewer #1 (Remarks to the Author):

Reviewer: Justin Filiberto

Lagain et al provide new crater count investigations to suggest that Tharsis and specifically two craters on Tharsis are the origin site for the depleted shergottites. The work is interesting and appropriate for Nature Communications. However, the accuracy of the writing needs improvement before it is ready for publication. I have included my comments directly in the pdf. Please do not hesitate to contact me with any questions. Some of the big picture comments:

1) How were the two craters chosen, especially crater 009? There are other craters with error bars that overlap closer to the data for the depleted shergottites than 009. The manuscript did not go through how it down selected to those craters.

2) Mantle potential discussion is not scientifically accurate with the publications. Please see my other papers that my Chemical Geology paper is based on (Filiberto et al., 2010; Filiberto and Dasgupta, 2011; Filiberto and Dasgupta, 2015), plus work by Carl Agee and Dave Draper in a series of papers (Agee and Draper, 2004; Draper, 2008), Musselwhite et al. (Musselwhite et al., 2006), and Kiefer et al. (Kiefer et al., 2015).

3) The discussion of the petrology of depleted shergottites is not accurate and is focused on the olivine-phyric shergottites only, but there are depleted shergottites with very different textures.

4) The discussion of a dry source region for the shergottites needs a reference. See: (Filiberto et al., 2016; McCubbin et al., 2016; Filiberto et al., 2019) (or others).

Agee, C.B., Draper, D.S., 2004. Experimental constraints on the origin of Martian meteorites and the composition of the Martian mantle. *Earth and Planetary Science Letters*, 224(3-4): 415-429.

Draper, D.S., 2008. Constraining the Depth of Martian Magma Ocean Crystallization: Role of Garnet Composition. *Lunar and Planetary Science*, XXXIX: Abstract #1313.

Filiberto, J., Dasgupta, R., 2011. Fe²⁺-Mg partitioning between olivine and basaltic melts: Applications to genesis of olivine-phyric shergottites and conditions of melting in the Martian interior. *Earth and Planetary Science Letters*, 304(3-4): 527-537.

Filiberto, J., Dasgupta, R., 2015. Constraints on the depth and thermal vigor of melting in the Martian mantle. *Journal of Geophysical Research: Planets*, 120(1): 2014JE004745.

Filiberto, J., Dasgupta, R., Kiefer, W.S., Treiman, A.H., 2010. High pressure, near-liquidus phase equilibria of the Home Plate basalt Fastball and melting in the Martian mantle. *Geophysical Research Letters*, 37(13): L13201, doi:10.1029/2010GL043999.

Filiberto, J., Gross, J., McCubbin, F.M., 2016. Constraints on the water, chlorine, and fluorine content of the Martian mantle. *Meteoritics & Planetary Science*, 51(11): 2023-2035.

Filiberto, J., McCubbin, F.M., Taylor, G.J., 2019. Chapter 2 - Volatiles in Martian Magmas and the Interior: Inputs of Volatiles Into the Crust and Atmosphere. In: Filiberto, J., Schwenzer, S.P. (Eds.), *Volatiles in the Martian Crust*. Elsevier, pp. 13-33.

Kiefer, W.S., Filiberto, J., Sandu, C., Li, Q., 2015. The effects of mantle composition on the peridotite solidus: Implications for the magmatic history of Mars. *Geochimica et Cosmochimica Acta*, 162(0): 247-258.

McCubbin, F.M. et al., 2016. Heterogeneous distribution of H₂O in the Martian interior: Implications for the abundance of H₂O in depleted and enriched mantle sources. *Meteoritics & Planetary Science*, 51(11): 2036-2060.

Musselwhite, D.S., Dalton, H.A., Kiefer, W.S., Treiman, A.H., 2006. Experimental petrology of the basaltic shergottite Yamato-980459: Implications for the thermal structure of the Martian mantle. *Meteoritics & Planetary Science*, 41(9): 1271-1290.

Reviewer #2 (Remarks to the Author):

Review of manuscript: "The Tharsis mantle source of depleted Shergottites revealed by 90 million impact craters" for *Nature Communications*, by Lagain et al.

This manuscript studies the crater source and source of the depleted shergottite martian meteorites using the novel Crater Detection Algorithm (CDA) and finds that the Tooting crater was the likely source crater of these rocks. This study is important as we currently do not know where the martian meteorite originate from at the martian surface. In addition, depleted shergottites are some of the most common martian samples. The CDA can also be used for other studies focusing on craters of different sizes. This manuscript is well-written and the implications are well-justified. I recommend publication after minor to moderate revisions.

General comments:

This manuscript needs to better explain the analyses that have been conducted in the main text in addition to the supplementary material; in particular it needs to be more explicit regarding the selection of the sizes of the studied craters and what crater (primary or secondary) are analyzed. It is also not clear how the 19 secondary ray systems were selected. See detailed comments for L73-115.

Detailed comments:

L 30. Add “for the depleted shergottite group” after 1.1 Myr ago

L31. Add “diameter” and 30 km and 20 km

L32. Shergottite is not a capitalized word. Change throughout the manuscript

L45. Shergottite groups are divided based on their rare earth element (REE) compositions but also their isotopic compositions (see Udry et al. 2020)

L48. It is unlikely that martian meteorites have undergone much crustal contamination that affected their REE, as it is not reflected in their major element compositions (see Udry et al. 2020). I recommend to remove “crustal contamination”

L50-52. The depleted group includes basaltic, olivine-phyric, and poikilitic shergottites: They have different textures (including poikilitic) and the basaltic shergottites do not contain olivine.

L53. Change “Northwest Africa (NWA)” to “NWA”

L73-74. What is the predicted size range of the primary crater source for depleted shergottites and thus what would be the size range of its secondary craters?

L81. Is the primary crater estimated to be <30km? If so, why?

L85. I suppose that 25 m and 300 m might be the possible size range for secondary craters. For someone who is not an expert in crater counting (as most of the readers of Nature Communications), it is not clear why only the size range of secondary craters and not also size range of primary crater is not also investigated. Please explain in the previous paragraph.

L.89. Add a space between Fig. and 1

L91-92. Is there a particular reason why the studied craters are divided into three size bins?

Figure 1. Include the color scale explanation in the caption. "Km" should be "km"

L104-105. It is not clear how the 19 craters were selected: is it based on the location of all the 25 m-300 m diameter secondary craters? Be more explicit regarding which sizes represent the primary craters versus secondary craters.

L108. Why are craters >7 km diameter separated from the others? Is it based on the 10 Myr date? Is it due to its 9% uncertainties?

Figure 2: In the legend, Tooting and 09-00015 should probably be the first two craters of the list. Explain in the caption what Tooting crater model correspond to the symbol with thin lines and with thick lines.

L140. "Crater" should not be capitalized

L158. 7 should be superscript

L166. Include a reference regarding the low alteration of depleted shergottites

L185-186. Add "crater" after "Gusev" and "Planum" after "Meridiani"

L414. Why only <150 m diameter craters were included to find the 19 impact craters when crater sizes up to 300 m were investigated?

Figure 3. It is not clear what the red dashed line represents

L511. Include Yoshizaki and McDonough (2020) as reference

Extended data fig 2. It might be because of the figure quality but it is difficult to distinguish the different colors

I hope the authors will find these reviews useful,

Arya Udry

References:

Udry, A., Howarth, G. H., Herd, C., Day, J. M. D., Lapen, T. J., & Filiberto, J. (2020). What martian meteorites reveal about the interior and surface of Mars. *Journal of Geophysical Research: Planets*, 55, <https://doi.org/10.1029/2020JE006523>.

Yoshizaki, Takashi, and William F. McDonough. "The composition of Mars." *Geochimica et Cosmochimica Acta* 273 (2020): 137-162.

Reviewer #3 (Remarks to the Author):

The authors indicate two impact craters in the Tharsis region on Mars as the most likely source of depleted Shergottites. For this, they identified rayed crater systems and considered the model ages of the primary craters and the surrounding ground. From the link between the Shergottites found on Earth and their proposed origin, the authors imply that the volcanic activity of the Tharsis region was

sourced deep in the mantle since the early Amazonian, sampling a depleted mantle, relic of Mars' early history.

The results of this paper are novel as it is the first time that the question regarding the potential origin of Martian meteorites is supported by a global analysis of secondary crater fields. The authors utilised a huge dataset of images and modern machine learning techniques to detect tens of millions of craters, which makes their investigation very thorough providing more evidence than ever. The findings of this work, from the source of Martian meteorites found on Earth to the implications for the interior and the evolution history of Mars, makes this work of interest for a very wide range of the planetary science community and the wider field.

Although the details provided in this paper do not suffice for the work to be reproducible, the study is well designed, and the conclusions are convincing. However, there is room for improvement in the evaluation of the automatic crater detection method. Although the development of the crater detection algorithm was published in another paper, the adaptation to this study was not evaluated in the most appropriate way. It is problematic in terms of design and estimates (see many of my comments).

The main issue with the evaluation concerns the crater sizes of interest. More specifically, the crater sizes considered in the confusion matrix and in the evaluation of the diameter estimation neither correspond to each other nor to various claims about the crater sizes of interest (from the text: <1 km, 25 - 300 m; from the evaluations: >60 m, 25 m - 2 km). According to the authors' performance acceptance limits (TPR>75%, FNR<25%, diameter estimation within 25% of the true diameter) and after some corrections of their confusion matrix' calculations, only craters with a diameter between 100 and 200 m are well detected both in terms of True Positive Rate and diameter estimation. These comprise only ~15% of the evaluation datapoints. I suggest that the acceptance limits be reconsidered, and the crater sizes of interest be well evaluated and justified. The limitations of the method as well as to what extent these affect the global density map upon which the results of this paper rely should be thoroughly discussed.

Overall, this paper is a great example of the potential that lies in using the most modern techniques and exploiting big datasets to answer questions that have been speculated about for years. This approach could, indeed should, be utilised in other topics of the planetary science and beyond.

Here are my comments, in order of appearance in the paper:

Abstract

"Here we developed a Crater Detection Algorithm that compiled a new database of 90 million impact craters to determine the launch position of Martian meteorites through the observation of secondary crater fields and consideration of the ejection age (1.1 Myr ago)."

The algorithm was not developed for this study. It was adapted to it. So, this should be clearer at the start of this sentence.

Main

“More than 280 pieces of 152 unique samples, originating from at least 11 source craters, are curated in the world’s collections¹.”

In the meantime (~7 months later), these are more than 300. I suggest updating this number accordingly.

“The temporary controversy regarding the significance of much older U-Pb ages for some of these samples now being settled⁸, the hypothesis that the Mojave impact crater, in old Noachian terrains, might be the source of all Shergottites⁵ can now be discarded.”

Do you mean “is now being settled”? Also, you could write out what U-Pb stands for, for those who are not familiar with the term.

“Identification of rayed craters is possible using images in the thermal infrared domain^{5,16}, but this approach is hampered by both image resolution (100 m/pixel), dust coverage (about half the Martian surface would remain not accessible by this technique) and physical properties of the impacted terrain.”

Could you refer to one or two of the physical properties that you mention? Also “both” is mostly used for 2 things, so here it would be better to erase it.

“Numerical simulations suggest that impact events capable of producing such fragments would form craters larger than ~3 km in diameter on the Martian surface⁹⁻¹⁰.”

“Secondary impacts reach a maximum size of about 2 to 5% of the primary crater diameter¹¹⁻¹³.”

“For instance, a 30 km crater would typically form secondaries smaller than 1 km diameter.”

“Existing databases of impact craters on Mars do not cover the range of diameters (< 1 km) relevant to find radial patterns of secondaries associated with primary craters < 30 km.”

“We developed an automated Crater Detection Algorithm (CDA)¹⁷, to identify craters between 25 m and 300 m in diameter across the entire surface of Mars.”

Since impact events capable of producing fragments with a velocity higher than the escape velocity, form craters larger than 3 km, and secondary impacts reach a maximum size of 2-5% of the primary crater (~60-150 m for a 3 km crater), and typically a 30 km crater forms secondaries of 1 km (actually up to 1,5 km) diameter, then why are you looking for craters between 25 and 300 m and not up to 1 km or 1,5 km? Where do the limits of 25 and 300 m come from and how do they affect your results?

“The algorithm was trained using High-Resolution Imagery System Experiment (HiRISE) images (25 cm/pixel) and applied on the global Context Camera (CTX) mosaic¹⁸ (5 m/pixel),”

What is the benefit of training the algorithm on a different dataset than the one that was used for the evaluation and application? Wouldn't the performance of the algorithm increase if it was trained on CTX data? Also, HiRISE stands for High Resolution Imaging Science Experiment.

“A color-coded crater density map is generated from this dataset (Fig.1) , with red, green, and blue channels corresponding to local crater density (in a 0.05° grid) for three diameter size ranges, in order of decreasing range of diameter, respectively $150 < D < 300$ m, $75 < D < 150$ m and < 75 m (see Methods and Extended Data Fig. 5).”

This density map or at least the Extended Data Figure 5 is missing a detailed legend on the units of the density. About how many craters (of each size class) are located in a $\sim 3 \times 3$ km grid when it is green for example? Furthermore, the rays of the two proposed primary craters are visualised as “bluish” and “greenish”. This means that the sizes of the majority of the therein detected secondary craters are below 75 m and 150 m respectively. The CDA algorithm overestimates the craters' sizes for craters smaller than 100 m up to 50% and this makes the map's selected diameter limits for the color-coding somewhat unreliable.

“19 secondary ray systems associated with large and recent primary craters are readily identified on this map (Fig. 1).”

A short explanation on how these were identified is of great importance.

“This is more than 1000 km farther than previously noted⁷.”

The “7” should be in superscript.

“Even though we cannot rule out 09-000015 crater as a plausible source for this group of meteorites, the geographic location of both craters allows us to link the source of depleted Shergottites to the Tharsis' dome magmatic activity and propose a geodynamic context for the largest volcanic edifice in the Solar System.”

Maybe change to “... depleted Shergottites to the Tharsis dome's ...”.

Methods

“Crater Detection Algorithm training.”

Is this section only describing the training phase?

“The key features are summarized here and followed by a description of the adaptation achieved for the purpose of this study.”

Maybe erase the “and”.

“To detect smaller impact craters and thus being able to identify rays of tens of meter-sized secondary craters, higher resolution datasets, such as HiRISE (High-Resolution Imagery System Experiment) and CTX (Context Camera) are necessary.”

“Rays of tens of meter-sized” gives the impression that you are looking for tens of craters with a diameter of approximately 1 m. Suggestion: “rays of sub-kilometre-sized”. Also, HiRISE stands for High Resolution Imaging Science Experiment.

“The high degree of detail found in these datasets makes the current training for the CDA inefficient due to the presence of 10s m-sized, pseudo-circular structures or other features within field dunes. Retraining the algorithm is therefore essential to accurately detect impact craters smaller than 1 km in diameter.”

Here again the “10s m-sized” is confusing. Also, I suggest extending this sentence to explain why exactly the pseudo-circular structures are a problem for the existing algorithm and how the retraining helped to solve it.

“Two significant hurdles were required for retraining: (1) increasing the speed of the training-evaluation step and (2) creating a new set of labelled images, i.e. a training library where all impact craters visible on a set of images are identified using HiRISE imagery⁴⁰.”

In general, machine learning methods include either a training and a test phase, or a training, a validation and a test phase. Thus, “training-evaluation step” could mean either training-test step or training-validation step. I think that here you mean training-test step. Validation usually refers to a part of the training phase (further tuning of the model parameters) like you mention below:

“The second challenge was achieved using the HiRISE mosaic⁴⁰ of the Jezero crater (E77-5-N18-0) where 2200 craters have been manually identified in the aim to train the CNN, of which 550 have been held out for validation.”),

whereas the test phase is for assessing the performance of the model. Please use these terms as correctly and clearly as possible throughout the paper to avoid any confusion of the reader.

Also, as commented previously, why did you train on HiRISE data and not on CTX and how do you think this affected your results?

“Two significant hurdles were required for retraining:”

“This is particularly critical when a few hundred gigabytes of imagery datasets (> 5 TB in the case of the CTX global mosaic¹⁸) must be analysed in a timely fashion.”

Based on this sentence this whole section refers not only to “retraining” like it is stated above nor only to the training phase of the algorithm as stated in the title of the section. I suggest you make this clear.

“To that end, we broke down all the processing steps of the previous version of the CDA17 to individual tasks, all with their own container that can be run individually and in parallel for different images. Each of which goes through all the individual steps producing crater locations in geographical coordinates of the original GeoTIFF image (using the spatial reference system specified therein).”

This part is confusing: Processing steps → individual tasks with their own container → each goes through all steps? So were the processing steps broken down to individual tasks at the end?

“Each HiRISE image is divided into tiles (960 x 960 pixels) before running the detection and analysed three times by the CNN (Convolutional Neural Network) at different resolutions. This ensures that larger craters that might be bigger than a single tile at native resolution are detected¹⁷.”

Could you please specify which HiRISE product was used, which three resolutions were selected and why? Also, mentioning that the image is first divided into tiles and then that it is analysed in three different resolutions is confusing. This should be a bit clearer like in the section 2.3.2 of the reference paper. Consider revising this sentence. Suggestion: “Before the detection, each HiRISE image is ...” followed by the steps in the right order.

“This labelled dataset was also augmented by applying a range of transformations (rotate, shear, scale, and translate) using YOLOv3⁽⁴¹⁾.”

How many samples were produced/used in total?

“The CTX global mosaic is the highest global image dataset currently available¹⁸. Its resolution of ~ 5 m/px allows us to identify 10s meter-sized impact craters.”

Here, again, the issue with the phrase “10s meter-sized” causes confusion.

“Each image is then downsampled in resolution from 40 to m/px to 160 m/px.”

Is the resolution of the mosaic 40 m/px and not 5 m/px? Or do you mean that the three resolutions mentioned before are between 40 and 160 m/px? This needs to be clearer.

“This step takes about 66 CPU hours but running in parallel reduces the computation time to generate the 15840 images to 45 minutes.”

Is this the total number of images of the CTX mosaic? You refer to it as if you have mentioned this number before, which is not the case. Also, maybe replace “to 45 minutes” to “in 45 minutes” or revise the sentence so that it is more grammatically correct.

“This set of images has been finally analysed by our retrained CNN in about 24 hours using 12 nodes from Pawsey Topaz cluster featuring dual Tesla V100 GPUs and another 3 hours for the NMS phase using one node from the Pawsey Magnus cluster, resulting in close to 95 million crater detections of which just under 90 million are smaller than 300 m in diameter.”

Does this mean that the algorithm also detects craters larger than 300 m? Please make this clear in the full length of your paper and explain why you refer to this specific number (300 m) in several places like in: “We developed an automated Crater Detection Algorithm (CDA)¹⁷, to identify craters between 25 m and 300 m in diameter across the entire surface of Mars.”

“To statistically validate the new results, we performed several counts on different types of terrains and compared the craters manually identified to the CDA’s detection.”

I suggest that you change “performed several counts” to “manually mapped craters”?

“Extended Data Table 1 shows how many craters have been manually identified for each terrain and for different diameter ranges. The number of true positive, false detection and missed craters are also indicated for each diameter range.”

This gives the impression that the table mainly shows the number of manually identified craters. I suggest merging the two sentences into one and emphasizing that the main purpose of the table is the performance assessment of the algorithm. Also, the number of craters that were manually mapped are over 60 m in diameter although you claim to have looked for craters from 25 m. The evaluation design should correspond to the diameters of interest for the study.

“Red boxes correspond to a poor true positive detection rate (< 75%) or a high false detection rate (> 25%) or a high missed craters rate (> 25%).”

Please, replace the first “or” with a comma. Also, consider using the official terms for these metrics: for example, TPR = True Positive Rate (without the word detection in between). The main issue here

and in the Extended Data Table 1 is that the calculation of what you call “false detection rate” is not an official metric. For this you divided the False Positives by the True Positives which does not give much information about the performance of the algorithm. I suggest erasing that part of the table and the text that refers to it. The false positive rate is the closest rate to what you calculated. However, it does not apply to this experiment because the “negative class” statistics cannot be completed. Normally, False Positive Rate = False Positives/Negatives and this study can only provide False Negatives. With these you can only calculate what you called “missed craters rate”: the False Negative Rate $FNR = 1 - TPR = FN/P$ like you have in the bottom part of the table. Two other metrics that would be informative for this study would be the precision = $TP/(TP+FP)$ and the F1 score. For all craters over 70m it seems that the precision is ~95% and the F1 score ~0.84, which show good performance of the algorithm.

“From this, the average true positive detection rate is higher than 75% for craters larger than 70m in diameter, ~ 85% for $D > 100$ m and higher than 90% for $D > 200$ m.”

In the Extended Data Table 1 you state that the True Positive Rate for craters larger than 70 m in diameter is 78,7%, but if we divide the True Positives by the Positives, the result is $530/706 * 100 = 75.1\%$. For $D > 100$ m: $283/371 * 100 = 76,3\%$ and for $D > 200$ m: $77/107 * 100 = 72\%$. For smaller diameters the calculations are wrong, too, as are for the FNR. The conclusion is that the algorithm achieves over 75% only for craters with a diameter between 70 and 200 m.

“The rate of false detection is always low (< 5%), whatever the considered diameter range.”

As I mentioned before this is not a real metric. There is no such thing as dividing False Positives by True Positives. Please remove this from the table and the text.

“Detections at high latitudes (> 45 degrees) are of the lowest quality compared to those closer to the equator, mostly due to the degraded morphology of impact craters.”

Is 45 degrees considered high or mid latitudes? Also, could you show an example of a degraded impact crater vs another impact crater to explain why the algorithm fails there? This does not seem to be the case in the previous paper about CDA, at least not up to 65 degrees of latitude.

“We observe an increase in the overestimation with the decrease in crater size due to the resolution of the image.”

Could you please comment on why you think the overestimation increases so much below 50 m of diameter? Are maybe 10 pixels in diameter approximately the limit for crater detection (see CDA reference paper as well)? Also, the largest crater that was manually mapped has a diameter of ~2 km. However, you claim to be interested in craters up to 300 m, so why did you test the method for craters as large as 2 km? This should be consistent and clear throughout the whole paper.

“Extended Data Fig. 3 shows a density plot version of the previous figure, showing the percentage of true positive detection falling into a difference bin of 5% and diameter of the ground truth bin of 10 m.”

This plot does not add a lot of information to the previous one. However, it shows that the evaluation of the method is actually based on a small subset of the manually detected craters. See also later comments on this. I suggest adding a figure with close ups of some good and bad detections.

“Our validation test shows that our database is statistically robust and complete from 100m in diameter.”

This is not well supported by your evaluation. From the Extended Data Table 1 and one of my previous comments, the craters over 200 m have been detected with a TPR of only 72% (<75% that is your acceptance rate), whereas the diameters estimation is within 25% of the manual detections only for craters over 100 m in size. So, the database is complete only for craters between 100 and 200 m. These are only 283 datapoints of the ~2000 craters that were tested. It is highly problematic that only ~15% of the evaluation datapoints are acceptable in both evaluations (TPR and diameter estimation).

“The CDA estimates crater diameter with a precision of ~25% for craters larger than 100m and is even better than that (within 15%) in more than 70% of cases.”

This should be clearly shown in the plots.

“While the counting accuracy decreases with decreasing diameter, true positive rates are equal to or better than average manual counting differences between two humans⁴² for craters larger than 70 m in diameter (Extended Data Table 1).”

Could you please add how much the differences between two humans are?

“Diameter estimation by the CDA was within 15 % (Extended Data Figs. 2 and 3) while the rate of false detection was less than ~5% across all terrain types for $D > 70$ m.”

This is not clear in the plots. Also, again, false detection rate is not a known metric.

“Degraded crater morphologies, mostly represented at high latitudes (> 45 degrees), are the main source of error in absolute counts and diameter estimation compared to those closer to the equator.”

This should be shown in some examples and/or references.

“We note that our catalog accounts for nearly all craters down to 70 – 100 m diameter range as shown on Extended Data Fig. 4.”

In the Y axis do you mean Number of Craters? In the Extended Data Fig. 4 the craters detected by CDA go up to 50 km in size. Does this include the detections from the previous paper, or did you detect here craters up this size? It should be clear which crater detections belong to which publication.

Reviewer #4 (Remarks to the Author):

This enthralling paper convincingly identifies one (or two) candidate source crater(s) for a group of martian meteorites—the depleted Shergottites—a major goal in planetary science since it was first discovered that rocks could be ejected from Mars and end up on Earth. As Tharsis, the biggest volcanic complex in the solar system, is the source region, this connection allows the authors to draw some important conclusions about the thermal state and evolution of Mars’ deep mantle for the first time.

The source crater is identified by a novel application of machine learning. The approximate size (~10 km) and age (~1 Ma) of the source crater has long been known. The challenge has been to identify it among the nearly 100,000 craters on Mars that are the right size. Here, the authors reduce that down to less than 20 candidates by identifying craters of the appropriate size that are surrounded by a field of so-called secondary craters. Secondary craters are formed by the impacts of abundant ejecta fragments thrown out of the primary crater (but not quite fast enough to escape Mars). Secondary crater fields are quickly eroded on Mars and are therefore indicative of a relatively recent impact. However, secondary craters associated with the most likely source craters of martian meteorites are 10s to 100s of meter in diameter, of which there are many, many millions on Mars. Far too many for a human to feasibly count.

The authors’ state-of-the-art crater detection algorithm has allowed them to use a high-performance computer to identify the vast majority of craters 100-m to 1-km in size (an astonishing 90 million), including the secondaries of candidate martian meteorite source craters. Crater density

maps made from these 90 millions craters reveal the 19 primary craters large enough and young enough to have ejected the Shergottites off Mars. The number and sizes of these candidates is almost exactly as expected for an ~8 Myr period on Mars, which leaves little chance that one or more candidate craters have been missed by this approach. A very comforting self-consistency check. When detailed crater counts on the ejecta blankets and the surrounding terrain are made to determine the age of the candidate craters, and the target rocks the craters were formed in, the 19 potential craters are reduced to one or two, both of which are located in the Tharsis province of Mars. Poetry.

The paper is well written and presented and is underpinned by careful attention to detail, expounded in the supporting materials. While I do have some comments and suggestions to improve the manuscript, these should not detract from my overall assessment that this is an excellent contribution well worthy of publication in a Nature-family journal. The multi-disciplinary nature of the research and novel application of machine learning make it perfectly suited to the broad readership of Nature Communications.

I have attached a Word copy of the MS with some comments and typos corrected. If this gets mangled by the manuscript portal, the authors are welcome to contact me for a copy. To those comments I add the following key points that I hope the authors will consider in revising their manuscript.

(1) A central assumption in the work is that the source rocks of the shergottite meteorites did not move between when they crystallised and when they were ejected from Mars. Is there evidence in the meteorites that precludes the rocks being transported from source region to another part of Mars by an impact? If so, it would be good to explain this; if not, I think this assumption needs to be explicitly stated, perhaps with some discussion as to how plausible this is. I note that most ejecta by volume is low or unshocked and cold; thus, while the rocks would be broken, they may not exhibit other metamorphic evidence of impact.

(2) I found the crater density colormaps in Fig. 1 and Extended Data Fig. 8 very challenging to interpret and contest the claim in the text that recent primary craters are “readily identified” in Fig. 1. I don’t dispute that the authors have indeed identified the secondary crater fields as claimed (they have examined these maps in much more detail than I can), and am very comforted by the self-consistency provided by subsequent crater counts. However, I do not find Fig. 1 to be convincing support for the identification of the candidate craters and in some cases the evidence in the close-up images of Fig. 8 is not very convincing, either (e.g., Canala, 19-000945, 16-001276). I would like to see the authors try to address this in the revised MS.

Now, I must admit to being among the ~8% of the male population that suffers from deuteranopia, but I have sought a second opinion that confirms that these plots are hard to interpret even for someone without my affliction. It is clear that considerable thought has gone into these plots, but I am not convinced that the cyclic RGB colourmap is an appropriate way to represent the data. It is

not an ideal choice for two reasons. First, it is not perceptually uniform, meaning that the eye is drawn to specific points on the scale (in this case cyan and yellow), meaning that pixels (and any features) with exactly that specific combination of crater densities will be more (artificially) obvious. Second, the underlying data is not cyclic—a portion of the colourwheel is not relevant, because you will never have small and large craters, but no medium craters. Whatever background the density map is overlaid onto is also potentially unhelpful; it's very difficult to tell what is background and what is crater density information. To address my concerns (and improve accessibility), I make the following suggestions.

a. Please consider an alternative way to plot the crater density data that uses a perceptually uniform colour scale. My tentative suggestion would be to try an approach that uses opacity (alpha) to depict total crater density and colour (on a perceptually uniform colour scale) to show some measure of the characteristic “size” of craters (the median crater size, perhaps?). Another suggestion would be to just plot the green channel. For Tooting, at least, I don't see much (if anything) more in the full colour image than I do in the green.

b. Regardless of the final approach settled on, the method used to determine the colour of each pixel in the figures based on the raw crater counts should be explained in the methods section. For example, ED Fig. 5 suggests to me that the different colour channels are either on or off depending on whether a certain crater density is exceeded. Is that correct? Or is the value of each channel proportional to crater density as suggested in the text? Also, have the crater densities been normalised to account for the fact that craters in the middle size range (green) are more abundant (globally) than the larger (red) or smaller size range (blue)? Is there an upper and/or lower bound on the crater density for which any colour is plotted? I assume so as there are no black regions. Finally, I am surprised that one of the channels is craters smaller than 75-m given the poor recall and precision of the detections below 70 m. Although I agree that this does not appear to be critical for the purpose of identifying the secondary crater fields, I think it should be explicitly acknowledged that the crater densities in this channel are very inaccurate.

c. It would be very helpful to include a legend in the images to help show how to interpret the colors in terms of density and size. And I would also consider adding annotations/labels (and or text in the captions) to draw readers attention to the radial patterns identified.

d. It may also be helpful to reconsider the background on which the crater density map is overlain.

(3) Regarding the crater counts of the 19 candidate mars meteorite source craters. Although this may appear obvious to cratering experts, I think it would be helpful to be more explicit about which craters are used to date the crater ejecta blankets and the surrounding terrain, for the benefit of general readers. In particular, I presume that these are not the same craters as have been used to identify the secondary crater fields. If they are, some explanation of how self-secondary craters (i.e., secondary craters on top of the ejecta blanket) are dealt with might be important.

(4) The purpose of the modelling shown in ED Fig. 9 is unclear and could be better linked to the story. The models are perfectly valid, but the only thing they appear to inform is the total volume of displaced material in the Tooting impact. If this number is significant, then it would be good to explain why. One potentially interesting observation might be to compare this volume to the total volume of ejecta fragments required to produce the secondary craters.

Congratulations to the authors for an excellent contribution that I hope to soon see in press.

Sincerely,

Gareth Collins

g.collins@imperial.ac.uk

Response to Reviewer #1

Dear Dr Filiberto,

Thank you very much for your constructive feedback on our manuscript. We have considered all the raised points and we address them below. Responses to each of your comment are highlighted in orange. We believe that your comments were very helpful in improving the accuracy and the readability of our manuscript. We also thank you for the list of references you added to the review, allowing us to get a decent overview of the literature in this field. Please see detailed responses below. All other comments raised in the PDF you attached to the review have been addressed (see tracked change version).

Reviewer: Justin Filiberto

Lagain et al. provide new crater count investigations to suggest that Tharsis and specifically two craters on Tharsis are the origin site for the depleted shergottites. The work is interesting and appropriate for Nature Communications. However, the accuracy of the writing needs improvement before it is ready for publication. I have included my comments directly in the pdf. Please do not hesitate to contact me with any questions. Some of the big picture comments:

1) How were the two craters chosen, especially crater 009? There are other craters with error bars that overlap closer to the data for the depleted shergottites than 009. The manuscript did not go through how it down selected to those craters.

Response: Fig. 2 is the key Figure to understand how the selection is achieved, which is based on crystallisation age and age of the surrounding terrain of each crater. The following text has been revised, and now reads:

“To further restrict the number of candidate crater sources for the depleted Shergottites launched 1.1 Myr ago, their crystallization ages (~ 330 Myr - ~ 570 Myr and ~ 2.4 Gyr³⁻⁴) are compared to the ages of terrains surrounding each crater. This approach assumes that crater counts on the area surrounding the crater source may provide the age of the volcanic episode comparable to the crystallization age of the meteorite. Surrounding terrain model ages have been derived for all impact craters using the recently revised Mars crater catalogue¹², compiled manually and complete down to 1 km¹¹⁻¹², completed for higher precision by manual mapping of smaller craters if necessary (Methods). Model ages obtained for different impact cratering rates, ranging between a factor of 2 around the reference impact flux over the last ~ 3 Gyr²² were also estimated (Methods, Fig. 2).”

2) Mantle potential discussion is not scientifically accurate with the publications. Please see my other papers that my Chemical Geology paper is based on (Filiberto et al., 2010; Filiberto and Dasgupta, 2011; Filiberto and Dasgupta, 2015), plus work by Carl Agee and Dave Draper in a series of papers (Agee and Draper, 2004; Draper, 2008), Musselwhite et al. (Musselwhite et al., 2006), and Kiefer et al. (Kiefer et al., 2015).

Response: We added some of the references you suggested in the manuscript and changed the text to make it more accurate, L. 203: “The high T_p inferred from ol-phyric depleted Shergottites defines an adiabatic gradient that crosses the Martian fertile mantle (unmelted mantle) solidus²⁹ at depth > 1000 km (Fig. 3A). This indicates that melting potentially started in the transition zone, marked by the appearance of γ -spinel (ringwoodite) or below in the lower mantle. However, direct comparison

between those Tp and a fertile mantle solidus, with Mg# of 0.75 – 0.77, can be tenuous as the latter would melt at lower temperature³³, with respect to a depleted mantle source characterised by higher Mg# of 0.85 – 0.86⁽³⁴⁾. Nevertheless, experimental results show that the composition of melt extracted from a fertile mantle at 8 and 10 GPa match the high magnesium content of some depleted shergottites³⁴, which is consistent with melting starting at least 10 GPa, i.e. at the bottom of the upper mantle.”.

3) The discussion of the petrology of depleted shergottites is not accurate and is focused on the olivine-phyric shergottites only, but there are depleted shergottites with very different textures.

Response: This is a very good point. We modified the text accordingly, L. 54: “This group includes basaltic and poikilitic textures but is largely dominated by olivine-phyric shergottites and is characterized by a common depletion in REE.”.

4) The discussion of a dry source region for the shergottites needs a reference. See: (Filiberto et al., 2016; McCubbin et al., 2016; Filiberto et al., 2019) (or others).

Response: The following reference has been added in the revised version of the manuscript: Filiberto et al., 2019.

Agee, C.B., Draper, D.S., 2004. Experimental constraints on the origin of Martian meteorites and the composition of the Martian mantle. *Earth and Planetary Science Letters*, 224(3-4): 415-429.

Draper, D.S., 2008. Constraining the Depth of Martian Magma Ocean Crystallization: Role of Garnet Composition. *Lunar and Planetary Science*, XXXIX: Abstract #1313.

Filiberto, J., Dasgupta, R., 2011. Fe²⁺-Mg partitioning between olivine and basaltic melts: Applications to genesis of olivine-phyric shergottites and conditions of melting in the Martian interior. *Earth and Planetary Science Letters*, 304(3-4): 527-537.

Filiberto, J., Dasgupta, R., 2015. Constraints on the depth and thermal vigor of melting in the Martian mantle. *Journal of Geophysical Research: Planets*, 120(1): 2014JE004745.

Filiberto, J., Dasgupta, R., Kiefer, W.S., Treiman, A.H., 2010. High pressure, near-liquidus phase equilibria of the Home Plate basalt Fastball and melting in the Martian mantle. *Geophysical Research Letters*, 37(13): L13201, doi:10.1029/2010GL043999.

Filiberto, J., Gross, J., McCubbin, F.M., 2016. Constraints on the water, chlorine, and fluorine content of the Martian mantle. *Meteoritics & Planetary Science*, 51(11): 2023-2035.

Filiberto, J., McCubbin, F.M., Taylor, G.J., 2019. Chapter 2 - Volatiles in Martian Magmas and the Interior: Inputs of Volatiles Into the Crust and Atmosphere. In: Filiberto, J., Schwenger, S.P. (Eds.), *Volatiles in the Martian Crust*. Elsevier, pp. 13-33.

Kiefer, W.S., Filiberto, J., Sandu, C., Li, Q., 2015. The effects of mantle composition on the peridotite solidus: Implications for the magmatic history of Mars. *Geochimica et Cosmochimica Acta*, 162(0): 247-258.

McCubbin, F.M. et al., 2016. Heterogeneous distribution of H₂O in the Martian interior: Implications for the abundance of H₂O in depleted and enriched mantle sources. *Meteoritics & Planetary Science*, 51(11): 2036-2060.

Musselwhite, D.S., Dalton, H.A., Kiefer, W.S., Treiman, A.H., 2006. Experimental petrology of the basaltic shergottite Yamato-980459: Implications for the thermal structure of the Martian mantle. *Meteoritics & Planetary Science*, 41(9): 1271-1290.

Response to Reviewer #2

Dear Dr Udry,

Thank you very much for your constructive feedback and your summary on our manuscript. We have considered all the raised points and we address them below. Responses to each of your comment are highlighted in cyan. We believe that your comments were very helpful in improving the readability and the accuracy of our manuscript. Please see detailed responses below.

Review of manuscript: “The Tharsis mantle source of depleted Shergottites revealed by 90 million impact craters” for Nature Communications, by Lagain et al.

This manuscript studies the crater source and source of the depleted shergottite martian meteorites using the novel Crater Detection Algorithm (CDA) and finds that the Tooting crater was the likely source crater of these rocks. This study is important as we currently do not know where the martian meteorite originate from at the martian surface. In addition, depleted shergottites are some of the most common martian samples. The CDA can also be used for other studies focusing on craters of different sizes. This manuscript is well-written and the implications are well-justified. I recommend publication after minor to moderate revisions.

General comments:

This manuscript needs to better explain the analyses that have been conducted in the main text in addition to the supplementary material; in particular it needs be more explicit regarding the selection of the sizes of the studied craters and what crater (primary or secondary) are analyzed. It is also not clear how the 19 secondary ray systems were selected. See detailed comments for L73-115.

Response: Other reviewers pointed this out. We clarified this in the manuscript: L. 97: “A careful survey of this map allowed the identification of 19 secondary ray systems associated with large and recent primary craters (Fig. 1).”, L. 112: “The ages of these 19 young impacts are determined using manual crater counts (excluding secondaries) on their ejecta blanket (see Methods)”, L. 130: “Surrounding terrain model ages have been derived for all impact craters using the recently revised Mars crater catalogue¹², compiled manually and complete down to 1 km¹¹⁻¹², completed for higher precision by manual mapping of smaller craters if necessary (Methods).” and more generally in the methods section: “Model age derivation and uncertainties.”.

Detailed comments:

L 30. Add “for the depleted shergottite group” after 1.1 Myr ago

Response: Thank you for pointing this out; we modified the text accordingly.

L31. Add “diameter” and 30 km and 20 km

Response: Thank you for pointing this out; we modified the text accordingly.

L32. Shergottite is not a capitalized word. Change throughout the manuscript

Response: Thank you for pointing this out; we modified the text accordingly.

L45. Shergottite groups are divided based on their rare earth element (REE) compositions but also their isotopic compositions (see Udry et al. 2020)

Response: Thank you for pointing this out; we added “and isotopic compositions”.

L48. It is unlikely that martian meteorites have undergone much crustal contamination that affected their REE, as it is not reflected in their major element compositions (see Udry et al. 2020). I recommend to remove “crustal contamination”

Response: Thank you for pointing this out; we removed “crustal contamination”.

L50-52. The depleted group includes basaltic, olivine-phyric, and poikilitic shergottites: They have different textures (including poikilitic) and the basaltic shergottites do not contain olivine.

Response: Thank you for pointing this out; this sentence is now: “This suggests that they were ejected as sub-meter rock boulders by a single meteoritic impact. This group includes basaltic, olivine-phyric and poikilitic textures and is characterized by a common depletion in REE.”

L53. Change “Northwest Africa (NWA)” to “NWA”

Response: Thank you for pointing this out; we modified the text accordingly.

L73-74. What is the predicted size range of the primary crater source for depleted shergottites and thus what would be the size range of its secondary craters?

Response: The only constraint on the crater diameter capable of ejecting material from the martian surface comes from impact modelling (Artemieva & Ivanov, 2009; Head et al, 2002) where the authors suggested a 3 km crater diameter as a minimum diameter corresponding to the energy threshold for ejection of rock fragments above the Martian escape velocity. Specific crater size constraints for a particular group of meteorites were attempted from the measure of the dwell time (time during which the meteorite experienced high-pressure condition) (e.g. Bowling et al., 2020). Scaling relationships between dwell time and impact size are derived in this study from numerical simulations (iSALE), but the authors recognize that the dwell time is also dependent of impact angle, impact velocity and material properties. This first attempt is an interesting step toward constraining the impactor size (and diameter) for each meteorite, and led to generally higher impact diameter than previous estimated (based on the assumption that the dwell time is simply related to impactor size and velocity). Considering these caveats, and the wide range of peak shock pressures have been reported for this group of meteorites, we believe that it is premature to use this parameter. For these reasons, we do not take this parameter into account to further restrict the crater size of the depleted shergottites source.

The relation between the size of a primary impact crater and the maximum size of its associated secondaries is given in the main text. Secondaries are assumed to be as large as 2 to 5% of the primary

crater (Robbins & Hynek, 2011, 2012; Lagain et al., 2021). Nevertheless, those estimates are based on complex impact craters. Extrapolation of those estimates for a simple crater of 3 km in diameter leads to 60 m-150 m as the size of the largest secondary crater but this is considered speculative.

L81. Is the primary crater estimated to be <30km? If so, why?

Response: The typical maximum size of secondaries associated with a 30 km crater lies between 600 m and 1.5 km for which not any global crater database covering this size range exists. We do not discard any primary craters larger than 30 km but give this size as an example to show the range of crater size we target to identify recent impact craters associated with secondaries.

L85. I suppose that 25 m and 300 m might be the possible size range for secondary craters. For someone who is not an expert in crater counting (as most of the readers of Nature Communications), it is not clear why only the size range of secondary craters and not also size range of primary crater is not also investigated. Please explain in the previous paragraph.

Response: To clarify this point, we modified the lines 74-75: "Thus, the crater source of depleted Shergottites, launched 1.1 Myr ago, should be associated with abundant secondaries in the ~10 - 300 m-size range." And lines 83-84: "We developed an automated Crater Detection Algorithm (CDA)17, to identify craters down to 25m in diameter across the entire surface of Mars."

L.89. Add a space between Fig. and 1

Response: Agreed and corrected.

L91-92. Is there a particular reason why the studied craters are divided into three size bins?

Response: It is important to visualize the secondary crater distributions in different size bins to be able to visualize all possible primary crater of different sizes. Dividing the secondary maps in different size bins is therefore achieved highlight secondary crater fields of primary craters of different sizes. The choice of dividing the map into three size bins is motivated by the fact that 3 is the maximum number to be able to make a color representation of the data (using RGB channels). The text has been modified to clarify the motivation of this choice: "To visualize secondaries, and therefore recent primary of different sizes, the density of secondaries for three ranges of sizes ($150 < D < 300$ m, $75 < D < 150$ m and < 75 m) are represented as a map (Fig. 1) using respectively the red, blue and channels (Methods and Supplementary Fig. 5)."

Figure 1. Include the color scale explanation in the caption. "Km" should be 'km"

Response: Agreed and corrected (see colour cube legend on Fig. 1).

L104-105. It is not clear how the 19 craters were selected: is it based on the location of all the 25 m-300 m diameter secondary craters? Be more explicit regarding which sizes represent the primary craters versus secondary craters.

Response: We clarified this in the main text and the methods:

L. 100: “. A careful survey of this map allowed the identification of 19 secondary ray systems associated with large and recent primary craters (Fig. 1).,”

L.115: The ages of these 19 young impacts are determined using manual crater counts (excluding secondaries) on their ejecta blanket (see Methods) and are used to test the completeness of the survey presented here”,

L. 129: “To further restrict the number of candidate crater sources for the depleted Shergottites launched 1.1 Myr ago, their crystallization ages (~330 Myr - ~570 Myr and ~2.4 Gyr³⁻⁴) are compared to the ages of terrains surrounding each crater. This approach assumes that crater counts on the area surrounding the crater source may provide the age of the volcanic episode comparable to the crystallization age of the meteorite. Surrounding terrain model ages have been derived for all impact craters using the recently revised Mars crater catalogue¹², compiled manually and complete down to 1 km¹¹⁻¹², completed for higher precision by manual mapping of smaller craters if necessary (Methods).”

And more generally in the methods section: “Model age derivation and uncertainties”

L108. Why are craters >7 km diameter separated from the others? Is it based on the 10 Myr date? Is it due to its 9% uncertainties?

Response: Among all craters identified as potential source of the Martian meteorites ejection younger than 10 Myr old, only one is smaller than 7 km: 20-003865 (D = 3 km). The size frequency distribution of the 17 other craters (i.e. those >7 km in diameter and younger than 10 Myr old) is consistent with the expected number of craters for this size range accumulated over the last 10 Myr on the entire surface of Mars (Hartmann, 2005). In other words, the 3 km crater is not considered in the isochrone fitting due to the lack of craters between 3 and 7 km in diameter. This is shown on ED Fig. 6.

Figure 2: In the legend, Tooting and 09-00015 should probably be the first two craters of the list. Explain in the caption what Tooting crater model correspond to the symbol with thin lines and with thick lines.

Response: Thank you for pointing this out; this was indeed not clear in the caption. We modified it accordingly: “Note that two ages are reported for Tooting, one (308 ± 41 Myr) corresponding to immediate surrounding (thin outline), and the other ($1.77^{+0.69}_{-0.58}$ Gyr) accounting for large craters around Tooting and partially filled-up by more recent lavas (thick outline). Craters in the legend are sorted by size (Extended Data Table 2).”

However, I disagree with the proposed order in the legend. Craters are sorted by size as they are on Extended Table 2.

L140. “Crater” should not be capitalized

Response: Sorry for the typo, I modified the text accordingly.

L158. 7 should be superscript

Response: Yes, sorry for the typo, I modified the text accordingly.

L166. Include a reference regarding the low alteration of depleted shergottites

Response: Another reviewer pointed the lack of clarity of this paragraph. It now reads: “Regarding the surface environment, analysis of volatiles in impact melt pockets of the Tissint meteorite shows post-magmatic and pre-impact low alteration by subsurface water which equilibrated with the present-day atmosphere²⁶.”

L185-186. Add “crater” after “Gusev” and “Planum” after “Meridiani”

Response: Thank you for pointing this out; we modified the text accordingly.

L414. Why only <150 m diameter craters were included to find the 19 impact craters when crater sizes up to 300 m were investigated?

Response: Sorry, this is an error, I fixed this in the revised version by replacing “150 m” by “300 m”.

Figure 3. It is not clear what the red dashed line represents

Response: This is now mentioned in the caption (melts pathways).

L511. Include Yoshizaki and McDonough (2020) as reference

Response: Thank you, this reference is now included in the revised version.

Extended data fig 2. It might because of the figure quality but it is difficult to distinguish the different colors

Response: Given the high density of points plotted on this figure, it has been chosen to use the transparency of markers to improve the readability of the figure. Although distinguishing the different colors can be difficult, the main point of this figure is that the CDA estimated the crater diameter within 25% of errors in the vast majority of craters located on all kind of terrains, except at high latitudes (cyan circles) where the CDA overestimated their diameter. Since cyan circles are easily distinguishable from the other colors and that this figure is completed by Extended Data Figure 3, we do not think it is necessary to change the symbology on Extended Data Figure 2. For your information, here is the same figure using plain colors:

I hope the authors will find these reviews useful,

Arya Udry

References:

Udry, A., Howarth, G. H., Herd, C., Day, J. M. D., Lapen, T. J., & Filiberto, J. (2020). What martian meteorites reveal about the interior and surface of Mars. *Journal of Geophysical Research: Planets*, 55, <https://doi.org/10.1029/2020JE006523>.

Yoshizaki, Takashi, and William F. McDonough. "The composition of Mars." *Geochimica et Cosmochimica Acta* 273 (2020): 137-162.

Response to Reviewer #3

Dear reviewer,

Thank you very much for your constructive feedback on our manuscript. We have considered all the raised points and we address them below. Responses to each of your comment are highlighted in green. We believe that your comments were very helpful in improving the readability and the accuracy of our manuscript and its presentation. Please see detailed responses below.

The authors indicate two impact craters in the Tharsis region on Mars as the most likely source of depleted Shergottites. For this, they identified rayed crater systems and considered the model ages of the primary craters and the surrounding ground. From the link between the Shergottites found on Earth and their proposed origin, the authors imply that the volcanic activity of the Tharsis region was sourced deep in the mantle since the early Amazonian, sampling a depleted mantle, relic of Mars' early history.

The results of this paper are novel as it is the first time that the question regarding the potential origin of Martian meteorites is supported by a global analysis of secondary crater fields. The authors utilised a huge dataset of images and modern machine learning techniques to detect tens of millions of craters, which makes their investigation very thorough providing more evidence than ever. The findings of this work, from the source of Martian meteorites found on Earth to the implications for the interior and the evolution history of Mars, makes this work of interest for a very wide range of the planetary science community and the wider field.

Although the details provided in this paper do not suffice for the work to be reproducible, the study is well designed, and the conclusions are convincing. However, there is room for improvement in the evaluation of the automatic crater detection method. Although the development of the crater detection algorithm was published in another paper, the adaptation to this study was not evaluated in the most appropriate way. It is problematic in terms of design and estimates (see many of my comments).

The main issue with the evaluation concerns the crater sizes of interest. More specifically, the crater sizes considered in the confusion matrix and in the evaluation of the diameter estimation neither correspond to each other nor to various claims about the crater sizes of interest (from the text: <1 km, 25 - 300 m; from the evaluations: >60 m, 25 m - 2 km). According to the authors' performance acceptance limits (TPR>75%, FNR<25%, diameter estimation within 25% of the true diameter) and after some corrections of their confusion matrix' calculations, only craters with a diameter between 100 and 200 m are well detected both in terms of True Positive Rate and diameter estimation. These comprise only ~15% of the evaluation datapoints. I suggest that the acceptance limits be reconsidered, and the crater sizes of interest be well evaluated and justified. The limitations of the method as well as to what extent these affect the global density map upon which the results of this paper rely should be thoroughly discussed.

Overall, this paper is a great example of the potential that lies in using the most modern techniques and exploiting big datasets to answer questions that have been speculated about for years. This approach could, indeed should, be utilised in other topics of the planetary science and beyond.

Response: Thank you for your constructive comments and kind conclusion. The major points you raised in your thorough review regarding the machine learning side of our study have been of great importance to improve the manuscript. The evaluation of the model is now performed using standard metrics (precision, recall, F1 score) and more details are provided regarding the current limitations of our dataset and its evaluation (see responses below). We are particularly grateful on the thorough review you provided to improve this part of the manuscript.

Here are my comments, in order of appearance in the paper:

Abstract

“Here we developed a Crater Detection Algorithm that compiled a new database of 90 million impact craters to determine the launch position of Martian meteorites through the observation of secondary crater fields and consideration of the ejection age (1.1 Myr ago).”

The algorithm was not developed for this study. It was adapted to it. So, this should be clearer at the start of this sentence.

Response: Agreed and corrected

Main

“More than 280 pieces of 152 unique samples, originating from at least 11 source craters, are curated in the world’s collections¹.”

In the meantime (~7 months later), these are more than 300. I suggest updating this number accordingly.

Response: Thank you for pointing this out; we updated the text accordingly as well as the reference.

“The temporary controversy regarding the significance of much older U-Pb ages for some of these samples now being settled⁸, the hypothesis that the Mojave impact crater, in old Noachian terrains, might be the source of all Shergottites⁵ can now be discarded.”

Do you mean “is now being settled”? Also, you could write out what U-Pb stands for, for those who are not familiar with the term.

Response: We clarified this paragraph: “The temporary controversy regarding the significance of much older Uranium - Plomb ages for some of these samples is now being settled⁸. The hypothesis according to which the Mojave impact crater, located in old Noachian terrains, might be the source of all Shergottites⁵ can now be discarded.”

“Identification of rayed craters is possible using images in the thermal infrared domain^{5,16}, but this approach is hampered by both image resolution (100 m/pixel), dust coverage (about half the Martian surface would remain not accessible by this technique) and physical properties of the impacted terrain.”

Could you refer to one or two of the physical properties that you mention? Also “both” is mostly used for 2 things, so here it would be better to erase it.

Response: The influence of the terrain’s physical properties on rays identification on thermal images is not well documented. We removed this part of the sentence.

“Numerical simulations suggest that impact events capable of producing such fragments would form craters larger than ~3 km in diameter on the Martian surface⁹⁻¹⁰.”

“Secondary impacts reach a maximum size of about 2 to 5% of the primary crater diameter¹¹⁻¹³.”

“For instance, a 30 km crater would typically form secondaries smaller than 1 km diameter.”

“Existing databases of impact craters on Mars do not cover the range of diameters (< 1 km) relevant to find radial patterns of secondaries associated with primary craters < 30 km.”

“We developed an automated Crater Detection Algorithm (CDA)¹⁷, to identify craters between 25 m and 300 m in diameter across the entire surface of Mars.”

Since impact events capable of producing fragments with a velocity higher than the escape velocity, form craters larger than 3 km, and secondary impacts reach a maximum size of 2-5% of the primary crater (~60-150 m for a 3 km crater), and typically a 30 km crater forms secondaries of 1 km (actually up to 1,5 km) diameter, then why are you looking for craters between 25 and 300 m and not up to 1 km or 1,5 km? Where do the limits of 25 and 300 m come from and how do they affect your results?

Response: This is a very good point. Identification of secondary crater fields formed by very young craters imply the presence of very small secondaries at the surface because their lifetime is very short due to the erosion. We describe this in the rest of the paragraph you quoted. We estimate that a secondary crater of 100m in diameter would be completely erased in 50 Myr. The presence of such secondaries around a larger crater is a direct proof of its freshness and allows to distinguish the very young large crater population from an older one. In the present study, we aim to locate large and very young impact craters. Analysing the distribution of craters smaller than 300 m is an arbitrary threshold but makes completely sense for our purpose. Regarding the lower boundary, 25 m, it is the smallest crater identifiable on a 5m/px image (i.e. CTX).

“The algorithm was trained using High-Resolution Imagery System Experiment (HiRISE) images (25 cm/pixel) and applied on the global Context Camera (CTX) mosaic¹⁸ (5 m/pixel),”

What is the benefit of training the algorithm on a different dataset than the one that was used for the evaluation and application? Wouldn't the performance of the algorithm increase if it was trained on CTX data?

Response: The current model is based on a retraining on HiRISE images and take advantage of the previous training performed on THEMIS imagery, presented in Benedix et al., 2020. When the current model (trained on HiRISE) evaluation has been performed we noticed that metrics were good enough for our purpose: detecting secondary crater rays where craters are as small as about 25 m on CTX imagery. There is no need to detect 100% of the craters to be able to identify rays of secondaries. We arbitrary defined a TPR of 75% as a satisfying threshold to achieve our goal. The benefit of training the network on HiRISE images (25cm/px) is that you can see on a single image more pseudo-circular structures than on a lower resolution image such as CTX. Thus the labelling of such images allowed us to train the model at distinguishing those structures that are not from an impact origin from true impact craters. Although those structures would be visible on CTX images, there are present at a larger spatial scale, which requires to label more images to get the same performance. This being time consuming, and being satisfied about the model performance, we decided to dedicate the investigation of a potential improvement of the model to another study. Nonetheless, a retraining on CTX images and in particular at high latitudes would certainly improve the current model in terms of precision and diameter estimation. However, the improvement of the recall would be limited as the number of small craters significantly decrease at mid and high latitudes (e.g. Kreslavsky & Head, 2018).

Also, HiRISE stands for High Resolution Imaging Science Experiment.

Response: Agreed and corrected

“A color-coded crater density map is generated from this dataset (Fig.1) , with red, green, and blue channels corresponding to local crater density (in a 0.05° grid) for three diameter size ranges, in order of decreasing range of diameter, respectively $150 < D < 300$ m, $75 < D < 150$ m and < 75 m (see Methods and Extended Data Fig. 5).”

This density map or at least the Extended Data Figure 5 is missing a detailed legend on the units of the density. About how many craters (of each size class) are located in a $\sim 3 \times 3$ km grid when it is green for example? Furthermore, the rays of the two proposed primary craters are visualised as “bluish” and “greenish”. This means that the sizes of the majority of the therein detected secondary craters are below 75 m and 150 m respectively. The CDA algorithm overestimates the craters’ sizes for craters smaller than 100 m up to 50% and this makes the map’s selected diameter limits for the color-coding somewhat unreliable.

Response: The lack of legend on this figure and the absence of rays in the 150m-300m diameter range has been extensively commented by the 4th reviewer. We have now addressed those issues by implementing a colour cube as a legend on the first figure and dedicated a section in the methods to the computation of this map. We also modified the Supplementary Fig. 5 to show the crater density for each size bin.

Also, considering the 150m-300m size range in the computation of the map brings an important information: the secondary crater size decrease with the distance to the associated primary crater (although beyond the scope of the present study and currently under investigation by our team, we are convinced that this is important to mention here). This is observed for Tooting but also for Kotka and 12-000685. This observation is consistent with impact modelling (Mc Ewen et al., 2005) and manual counts (Quantin et al., 2016) and further demonstrates that even if there are systematic errors in the diameter estimation of detected craters by the CDA, those ones are overall conservative between one crater and another, at least for $D > 50$ m (see kernel density estimation on ED Fig 3).

Regarding errors on the diameter estimation by our CDA, we acknowledge them both in the method section and the main text. All measurements are submitted to errors/uncertainties, even manual crater mapping that has been used to estimate the performance of the CDA. The diameter estimation errors derived here is largely acceptable for the purpose of our study since they have negligible consequences on recognizing the main systems of secondary craters. Nevertheless, we agree that those uncertainties are essential to mention in the main text as well as in the caption of Figure 1 so the readers would not miss this important information.

“19 secondary ray systems associated with large and recent primary craters are readily identified on this map (Fig. 1).”

A short explanation on how these were identified is of great importance.

Response: The identification of these craters has been performed by a careful survey using the crater density map. This is a common task, performed in many studies such as Werner et al. (2014), Tornabene et al. (2006, 2012), Robbins & Hynes (2012) and Lagain et al., (2021), which does not deserve further explanations. However, it needs to be clarified in the text: L.93: “A careful visualisation of this map allowed the identification of 19 secondary ray systems associated with large and recent primary craters (Fig. 1).”

“This is more than 1000 km farther than previously noted⁷.”

The “7” should be in superscript.

Response: Agreed and corrected.

“Even though we cannot rule out 09-000015 crater as a plausible source for this group of meteorites, the geographic location of both craters allows us to link the source of depleted Shergottites to the

Tharsis' dome magmatic activity and propose a geodynamic context for the largest volcanic edifice in the Solar System.”

Maybe change to “... depleted Shergottites to the Tharsis dome's ...”.

Response: Agreed and corrected.

Methods

“Crater Detection Algorithm training.”

Is this section only describing the training phase?

Response: We generalized the title of the section as some parts of it are beyond the training: “The Crater Detection Algorithm”

“The key features are summarized here and followed by a description of the adaptation achieved for the purpose of this study.”

Maybe erase the “and”.

Response: Agreed and corrected

“To detect smaller impact craters and thus being able to identify rays of tens of meter-sized secondary craters, higher resolution datasets, such as HiRISE (High-Resolution Imagery System Experiment) and CTX (Context Camera) are necessary.”

“Rays of tens of meter-sized” gives the impression that you are looking for tens of craters with a diameter of approximately 1 m. Suggestion: “rays of sub-kilometre-sized”. Also, HiRISE stands for High Resolution Imaging Science Experiment.

Response: Agreed and corrected

“The high degree of detail found in these datasets makes the current training for the CDA inefficient due to the presence of 10s m-sized, pseudo-circular structures or other features within field dunes. Retraining the algorithm is therefore essential to accurately detect impact craters smaller than 1 km in diameter.”

Here again the “10s m-sized” is confusing. Also, I suggest extending this sentence to explain why exactly the pseudo-circular structures are a problem for the existing algorithm and how the retraining helped to solve it.

Response: We clarified the sentence accordingly, L.332: “More specifically, landscapes within field dunes exhibit circular features related to erosion were essential to include in the training dataset to avoid detecting them.”

“Two significant hurdles were required for retraining: (1) increasing the speed of the training-evaluation step and (2) creating a new set of labelled images, i.e. a training library where all impact craters visible on a set of images are identified using HiRISE imagery⁴⁰.”

In general, machine learning methods include either a training and a test phase, or a training, a validation and a test phase. Thus, “training-evaluation step” could mean either training-test step or training-validation step. I think that here you mean training-test step. Validation usually refers to a part of the training phase (further tuning of the model parameters) like you mention below:

“The second challenge was achieved using the HiRISE mosaic⁴⁰ of the Jezero crater (E77-5-N18-0) where 2200 craters have been manually identified in the aim to train the CNN, of which 550 have been held out for validation.”),

whereas the test phase is for assessing the performance of the model. Please use these terms as correctly and clearly as possible throughout the paper to avoid any confusion of the reader.

Response: Thank you for pointing this out. Thanks to your comment, I learn myself that “validation” and “test” terms are often reversed in the literature leading to confusion within this community. We modified this section to clarify this.

Also, as commented previously, why did you train on HiRISE data and not on CTX and how do you think this affected your results?

Response: See response above

“Two significant hurdles were required for retraining:”

“This is particularly critical when a few hundred gigabytes of imagery datasets (> 5 TB in the case of the CTX global mosaic¹⁸) must be analysed in a timely fashion.”

Based on this sentence this whole section refers not only to “retraining” like it is stated above nor only to the training phase of the algorithm as stated in the title of the section. I suggest you make this clear.

Response: This is correct and we changed the title of this section accordingly (see response above)

“To that end, we broke down all the processing steps of the previous version of the CDA17 to individual tasks, all with their own container that can be run individually and in parallel for different images. Each of which goes through all the individual steps producing crater locations in geographical coordinates of the original GeoTIFF image (using the spatial reference system specified therein).”

This part is confusing: Processing steps ? individual tasks with their own container ? each goes through all steps? So were the processing steps broken down to individual tasks at the end?

Response: I agree that this part was a confusing. We have clarified the presentation of these processing steps, L. 278: “To that end, we broke down all the processing steps (image reprojection and downsampling, tiling, scoring) of the previous version of the CDA17 to individual tasks, all with their own container. Thus, each of the task that can be run individually and in parallel for different images.”.

“Each HiRISE image is divided into tiles (960 x 960 pixels) before running the detection and analysed three times by the CNN (Convolutional Neural Network) at different resolutions. This ensures that larger craters that might be bigger than a single tile at native resolution are detected¹⁷.”

Could you please specify which HiRISE product was used, which three resolutions were selected and why? Also, mentioning that the image is first divided into tiles and then that it is analysed in three

different resolutions is confusing. This should be a bit clearer like in the section 2.3.2 of the reference paper. Consider revising this sentence. Suggestion: “Before the detection, each HiRISE image is ...” followed by the steps in the right order.

Response: This part of the section explains the pipeline in general and the image used for training was originally mentioned at the end of the section (E77-5-N18-0). The order of the steps were indeed confusing. We are now presenting the training before the modifications performed on the pipeline (i.e. challenges (1) and (2)). The scoring on two downsampling images and on the raw resolution image are now mentioned and we revised the sentence as suggested.

“This labelled dataset was also augmented by applying a range of transformations (rotate, shear, scale, and translate) using YOLOv3(41).”

How many samples were produced/used in total?

Response: The repository where the samples were stored experienced an unexpected purge in April 2021. Unfortunately, I am afraid we are not able to retrieve this information anymore. Also, for internal reasons related to access to the supercomputer resources, we are not able to reproduce the augmentation to get this information at the time of writing. Nevertheless, the CNN was trained on 300 epochs over 75% of the marked tiles, which gives about 200,000 samples for the augmented dataset in total.

“The CTX global mosaic is the highest global image dataset currently available¹⁸. Its resolution of ~ 5 m/px allows us to identify 10s meter-sized impact craters.”

Here, again, the issue with the phrase “10s meter-sized” causes confusion.

Response: Agreed and corrected, L.293: “Its resolution of ~ 5 m/px allows us to identify impact craters as small as 25 m.”

“Each image is then downsampled in resolution from 40 to m/px to 160 m/px.”

Is the resolution of the mosaic 40 m/px and not 5 m/px? Or do you mean that the three resolutions mentioned before are between 40 and 160 m/px? This needs to be clearer.

Response: This point needs indeed to be clarified. The resolution of the mosaic is 5m/px. The images were downsampled at lower resolutions of 40 m/px and 160m/px, thus producing a set of 3 images at 3 different resolution for each tile of the mosaic. I clarified this point L.366:” Each image is then downsampled in resolution from the raw resolution (5 m/px) to 40 to m/px and 160 m/px.”

“This step takes about 66 CPU hours but running in parallel reduces the computation time to generate the 15840 images to 45 minutes.”

Is this the total number of images of the CTX mosaic? You refer to it as if you have mentioned this number before, which is not the case. Also, maybe replace “to 45 minutes” to “in 45 minutes” or revise the sentence so that it is more grammatically correct.

Response: Agreed and corrected: L.368:“This step takes about 66 CPU hours but running in parallel reduces the computation time to generate the 15840 images that compose the entire CTX mosaic in 45 minutes.”

“This set of images has been finally analysed by our retrained CNN in about 24 hours using 12 nodes from Pawsey Topaz cluster featuring dual Tesla V100 GPUs and another 3 hours for the NMS phase

using one node from the Pawsey Magnus cluster, resulting in close to 95 million crater detections of which just under 90 million are smaller than 300 m in diameter.”

Does this mean that the algorithm also detects craters larger than 300 m? Please make this clear in the full length of your paper and explain why you refer to this specific number (300 m) in several places like in: “We developed an automated Crater Detection Algorithm (CDA)17, to identify craters between 25 m and 300 m in diameter across the entire surface of Mars.”

Response: This is a very good point, we need to be consistent with this all along the paper. We clarified this in the revised version. For example, L. 96: “In this work, we do not analyse the distribution of detections > 300 m as the presence of secondaries above that size are not a discriminant factor for the freshness of their associated large primary crater.” and L.305: “This results in 94,773,450 crater detections, of which just 89,054,458 are smaller than 300 m in diameter and used to analyse the spatial crater distribution presented on Fig. 1.”,

“To statistically validate the new results, we performed several counts on different types of terrains and compared the craters manually identified to the CDA’s detection.”

I suggest that you change “performed several counts” to “manually mapped craters”?

Response: Agreed and corrected.

“Extended Data Table 1 shows how many craters have been manually identified for each terrain and for different diameter ranges. The number of true positive, false detection and missed craters are also indicated for each diameter range.”

This gives the impression that the table mainly shows the number of manually identified craters. I suggest merging the two sentences into one and emphasizing that the main purpose of the table is the performance assessment of the algorithm. Also, the number of craters that were manually mapped are over 60 m in diameter although you claim to have looked for craters from 25 m. The evaluation design should correspond to the diameters of interest for the study.

Response: Regarding the first point of your comment, we merged both sentences into one to make it clearer. I partly disagree with the second point of your comment. In ED Table 1, the line “P” mentions the total number of manually mapped craters, including those below 60m. For volcanic terrains, 364 craters are smaller than 60 m ($437-73=364$). We evaluated the whole population of craters in the manual counting against the CDA results on lines “TP”, “FP” and “FN”. We did not break down the evaluation below 60 meters for two main reasons: manual counting below this size becomes too uncertain in terms of (1) completeness and (2) interpretation of their impact origin, mostly due to the resolution of the image. Thus, they cannot be considered as a “ground truth” with high confidence. Although, we use this size range to compute the crater density map shown on Fig.1 and identify candidate crater source for the ejection of Martian meteorites, craters in the 25-60m size range are approximately as numerous (16,273,747) as the craters in the 60m-75m size range (16,615,952) (see ED Fig.4). Discarding this size range would tend to lose precious information regarding the crater rays and evaluating those detections per diameter range on the same area we already consider is tricky as mentioned above (point (1) and (2)).

“Red boxes correspond to a poor true positive detection rate ($< 75\%$) or a high false detection rate ($> 25\%$) or a high missed craters rate ($> 25\%$).”

Please, replace the first “or” with a comma. Also, consider using the official terms for these metrics: for example, TPR = True Positive Rate (without the word detection in between). The main issue here and in the Extended Data Table 1 is that the calculation of what you call “false detection rate” is not an official metric. For this you divided the False Positives by the True Positives which does not give much information about the performance of the algorithm. I suggest erasing that part of the table and the text that refers to it. The false positive rate is the closest rate to what you calculated. However, it does not apply to this experiment because the “negative class” statistics cannot be completed. Normally, False Positive Rate = False Positives/Negatives and this study can only provide False Negatives. With these you can only calculate what you called “missed craters rate”: the False Negative Rate $FNR = 1 - TPR = FN/P$ like you have in the bottom part of the table. Two other metrics that would be informative for this study would be the precision = $TP/(TP+FP)$ and the F1 score. For all craters over 70m it seems that the precision is ~95% and the F1 score ~0.84, which show good performance of the algorithm.

Response: Thank you for this constructive comment. We wanted to make this evaluation as much accessible as possible to a wide community of readers. We simplified too much this part of the methods and neglected the accuracy and validity of the evaluation presentation. We modified this part and the table accordingly. We replaced all metrics previously used by the recall, the precision and the F1 score. Also, regarding the metrics computed by taking into account all types of geological units (the last column), the fraction area represented by each type is not the same on the whole surface of Mars and some overlap (high latitudes is a particular case in the lowlands for instance and apron units corresponds to a very small fraction of the whole surface). Although related metrics are mentioned in the table, it does not make sense to qualify (using the proposed color code (red/green)) the overall performance of the CDA based on this. This is now mentioned in the manuscript and the whole section has been rewritten following your recommendations.

“From this, the average true positive detection rate is higher than 75% for craters larger than 70m in diameter, ~ 85% for $D > 100$ m and higher than 90% for $D > 200$ m.”

In the Extended Data Table 1 you state that the True Positive Rate for craters larger than 70 m in diameter is 78,7%, but if we divide the True Positives by the Positives, the result is $530/706 * 100 = 75.1\%$. For $D > 100$ m: $283/371 * 100 = 76,3\%$ and for $D > 200$ m: $77/107 * 100 = 72\%$. For smaller diameters the calculations are wrong, too, as are for the FNR. The conclusion is that the algorithm achieves over 75% only for craters with a diameter between 70 and 200 m.

Response: Agree and corrected (see comment above)

“The rate of false detection is always low (< 5%), whatever the considered diameter range.”

As I mentioned before this is not a real metric. There is no such thing as dividing False Positives by True Positives. Please remove this from the table and the text.

Response: Agree and corrected

“Detections at high latitudes (> 45 degrees) are of the lowest quality compared to those closer to the equator, mostly due to the degraded morphology of impact craters.”

Is 45 degrees considered high or mid latitudes? Also, could you show an example of a degraded impact crater vs another impact crater to explain why the algorithm fails there? This does not seem to be the case in the previous paper about CDA, at least not up to 65 degrees of latitude.

Response: High latitudes correspond to latitudes above 55 degrees. Nevertheless, we referred here to areas where periglacial processes (leading to high erosion rate) are active on Mars, which is about >45 degrees (and lower on some lowland provinces). We corrected this in the text: L.321: “Detections at mid and high latitudes (> 50°) are of the lowest quality compared to those closer to the equator, mostly due to the degraded morphology of impact craters.”.

Below is an example of two degraded craters located at 52 degrees of latitudes (background: CTX mosaic). Red circles are manually mapped craters while the green box corresponds to a CDA detection (~500m). On the right panel, the smallest crater seems to exhibit more pronounced rims (in particular at the NE) and is slightly less degraded. This can explain why it has been picked by our CDA and not the large one.

“We observe an increase in the overestimation with the decrease in crater size due to the resolution of the image.”

Could you please comment on why you think the overestimation increases so much below 50 m of diameter? Are maybe 10 pixels in diameter approximately the limit for crater detection (see CDA reference paper as well)? Also, the largest crater that was manually mapped has a diameter of ~2 km. However, you claim to be interested in craters up to 300 m, so why did you test the method for craters as large as 2 km? This should be consistent and clear throughout the whole paper.

Response: The resolution of the image is clearly responsible of this trend. This was stated in the submitted manuscript (L.391: “We observe an increase in the overestimation with the decrease in crater size due to the resolution of the image.”). We also added the following sentence: “This is especially true for craters < 50 m.”

We tested the performance of the CDA for a wider range of craters before selecting the crater size thresholds from which the crater density map would be used to identify rays of secondaries. Including a wider range of crater size in this evaluation is of important for future studies based on this database.

“Extended Data Fig. 3 shows a density plot version of the previous figure, showing the percentage of true positive detection falling into a difference bin of 5% and diameter of the ground truth bin of 10 m.”

This plot does not add a lot of information to the previous one. However, it shows that the evaluation of the method is actually based on a small subset of the manually detected craters. See also later comments on this. I suggest adding a figure with close ups of some good and bad detections.

Response: We modified the figure and the caption accordingly:

“Extended Data Figure 3. Kernel density estimator of the data presented on Extended Data Fig. 2 (shown as black crosses here) showing that detections larger than 100m are estimated within 25% of uncertainties compared to manually identified impact craters. Close-ups illustrate the detections seen in CTX (in blue) compared to their manual counterpart (in red). Readers are referred to the X-axis for the scale of each image.”

“Our validation test shows that our database is statistically robust and complete from 100m in diameter.”

This is not well supported by your evaluation. From the Extended Data Table 1 and one of my previous comments, the craters over 200 m have been detected with a TPR of only 72% (<75% that is your acceptance rate), whereas the diameters estimation is within 25% of the manual detections only for craters over 100 m in size. So, the database is complete only for craters between 100 and 200 m. These are only 283 datapoints of the ~2000 craters that were tested. It is highly problematic that only ~15% of the evaluation datapoints are acceptable in both evaluations (TPR and diameter estimation).

Response: I agree, this needs to be acknowledged and deserves an extensive evaluation of the data for $D > 200\text{m}$. This is beyond the scope of our study and left for future investigations as this has not any consequence on our interpretations and conclusions. The whole section has been rewritten with a better description of the current limitations and the tone has been tuned down regarding the robustness of the database: e.g. L. 342: “The CDA estimates crater diameter with a precision of ~25% for craters larger than 100m. While the counting accuracy decreases with decreasing diameter (in particular close to 10x the native resolution of the image, i.e. 50m), true positive rates are equal to or better than average manual counting differences between two humans⁴⁵ ($\sim \pm 30\%$) for craters larger than 70 m in diameter (Supplementary Table 1). Degraded crater morphologies, mostly represented at mid and high latitudes (> 45 degrees), are the main source of error in absolute counts and diameter estimation compared to those closer to the equator, as described in the work presented in a previous version of our algorithm¹⁷. Also, while the number of detections in the 25 m – 75 m is 32,889,699 (corresponding to one third of the total detections), the number of craters detected within the higher bracket of this range (60 m – 75 m, $N = 16,615,952$) is approximately equal to those smaller than 60 m ($N = 16,273,747$) (see Supplementary Fig. 4 for a plot of the Crater Size-Frequency Distribution of the detections). The population of small craters < 60 m, where the diameter is generally overestimated up to 50% does not

dominate the population of craters used to populate the blue band of the density map shown on Fig. 1.” And L. 358: “since craters > 200 m constitutes a minor portion of the detections tested in the present work, an extensive evaluation of larger craters is needed and left for future investigation. It is of note that the usual caveats and limitations of an automatic algorithm, including proportion of false positives, have negligible consequences on recognizing the main systems of secondary craters. Given the large numbers of secondaries associated with a given primary, a complete dataset is not required when using the CDA as a visualization tool.”

“The CDA estimates crater diameter with a precision of ~25% for craters larger than 100m and is even better than that (within 15%) in more than 70% of cases.”

This should be clearly shown in the plots.

Response: The last part of this sentence has been removed. The precision of 25% in the diameter estimation is however shown with dashed lines on ED Fig.2 and with the KDE on ED Fig.3.

“While the counting accuracy decreases with decreasing diameter, true positive rates are equal to or better than average manual counting differences between two humans⁴² for craters larger than 70 m in diameter (Extended Data Table 1).”

Could you please add how much the differences between two humans are?

Response: This is detailed in Robbins et al. (2014) (<http://dx.doi.org/10.1016/j.icarus.2014.02.022>) and approximately equal to 30% (depending on mapping interface, dataset, expertise degree...). This is now mentioned L. 399.

“Diameter estimation by the CDA was within 15 % (Extended Data Figs. 2 and 3) while the rate of false detection was less than ~5% across all terrain types for $D > 70$ m.”

This is not clear in the plots. Also, again, false detection rate is not a known metric.

Response: We modified this sentence accordingly: L. 402: “Diameter estimation by the CDA was within 25 % for $D > 100$ m (Extended Data Figs. 2 and 3).” Regarding the plots, this has been addressed above.

“Degraded crater morphologies, mostly represented at high latitudes (> 45 degrees), are the main source of error in absolute counts and diameter estimation compared to those closer to the equator.”

This should be shown in some examples and/or references.

Response: This effect has been presented in details in our previous paper (Benedix et al., 2020). We are now citing this article at the end of the sentence.

“We note that our catalog accounts for nearly all craters down to 70 – 100 m diameter range as shown on Extended Data Fig. 4.”

In the Y axis do you mean Number of Craters? In the Extended Data Fig. 4 the craters detected by CDA go up to 50 km in size. Does this include the detections from the previous paper, or did you detect here craters up this size? It should be clear which crater detections belong to which publication.

Response: All detections in the red histogram have been detected in this work only, on CTX imagery, none come from Benedix et al., 2020. We clarified this in the legend and caption of the figure. The Y axis has been changed to “Number of craters”.

Response to Reviewer #4

Dear Dr Collins,

Thank you very much for your constructive feedback on our manuscript. We have considered all the raised points and we address them below. Responses to each of your comment are highlighted in purple. We believe that your comments were very helpful in improving the readability and the accuracy of our manuscript. Please see detailed responses below.

This enthralling paper convincingly identifies one (or two) candidate source crater(s) for a group of martian meteorites—the depleted Shergottites—a major goal in planetary science since it was first discovered that rocks could be ejected from Mars and end up on Earth. As Tharsis, the biggest volcanic complex in the solar system, is the source region, this connection allows the authors to draw some important conclusions about the thermal state and evolution of Mars' deep mantle for the first time.

The source crater is identified by a novel application of machine learning. The approximate size (~10 km) and age (~1 Ma) of the source crater has long been known. The challenge has been to identify it among the nearly 100,000 craters on Mars that are the right size. Here, the authors reduce that down to less than 20 candidates by identifying craters of the appropriate size that are surrounded by a field of so-called secondary craters. Secondary craters are formed by the impacts of abundant ejecta fragments thrown out of the primary crater (but not quite fast enough to escape Mars). Secondary crater fields are quickly eroded on Mars and are therefore indicative of a relatively recent impact. However, secondary craters associated with the most likely source craters of martian meteorites are 10s to 100s of meter in diameter, of which there are many, many millions on Mars. Far too many for a human to feasibly count.

The authors' state-of-the-art crater detection algorithm has allowed them to use a high-performance computer to identify the vast majority of craters 100-m to 1-km in size (an astonishing 90 million), including the secondaries of candidate martian meteorite source craters. Crater density maps made from these 90 millions craters reveal the 19 primary craters large enough and young enough to have ejected the Shergottites off Mars. The number and sizes of these candidates is almost exactly as expected for an ~8 Myr period on Mars, which leaves little chance that one or more candidate craters have been missed by this approach. A very comforting self-consistency check. When detailed crater counts on the ejecta blankets and the surrounding terrain are made to determine the age of the candidate craters, and the target rocks the craters were formed in, the 19 potential craters are reduced to one or two, both of which are located in the Tharsis province of Mars. Poetry.

The paper is well written and presented and is underpinned by careful attention to detail, expounded in the supporting materials. While I do have some comments and suggestions to improve the manuscript, these should not detract from my overall assessment that this is an excellent contribution well worthy of publication in a Nature-family journal. The multi-disciplinary nature of the research and novel application of machine learning make it perfectly suited to the broad readership of Nature Communications.

Response: Thank you for your kind comment and summary. It is always a pleasure to see the multi-disciplinary value of our work recognised by other members of the community!

I have attached a Word copy of the MS with some comments and typos corrected. If this gets mangled by the manuscript portal, the authors are welcome to contact me for a copy. To those comments I add the following key points that I hope the authors will consider in revising their manuscript.

Response: All comments and suggestions raised in the word document attached to the review have been addressed.

(1) A central assumption in the work is that the source rocks of the shergottite meteorites did not move between when they crystallised and when they were ejected from Mars. Is there evidence in the meteorites that precludes the rocks being transported from source region to another part of Mars by an impact? If so, it would be good to explain this; if not, I think this assumption needs to be explicitly stated, perhaps with some discussion as to how plausible this is. I note that most ejecta by volume is low or unshocked and cold; thus, while the rocks would be broken, they may not exhibit other metamorphic evidence of impact.

Response: This is a good question. It would be always possible that a rocks sampled at a given place by a meteoritic impact had been previously transported by other processes (fluvial/glacial transport, previous impact with no evidence of shock). We will assume here that the rocks were generated at the impact location by magmatic processes.

This is now explicitly stated L. 53: “Although it would be possible that a rock sampled at a given place by a meteoritic impact had been previously transported by other processes (such as a previous impact with no evidence of shock), we will assume here that the rocks were generated at the impact location by magmatic processes.”

(2) I found the crater density colormaps in Fig. 1 and Extended Data Fig. 8 very challenging to interpret and contest the claim in the text that recent primary craters are “readily identified” in Fig. 1. I don’t dispute that the authors have indeed identified the secondary crater fields as claimed (they have examined these maps in much more detail than I can), and am very comforted by the self-consistency provided by subsequent crater counts. However, I do not find Fig. 1 to be convincing support for the identification of the candidate craters and in some cases the evidence in the close-up images of Fig. 8 is not very convincing, either (e.g., Canala, 19-000945, 16-001276). I would like to see the authors try to address this in the revised MS.

Now, I must admit to being among the ~8% of the male population that suffers from deuteranopia, but I have sought a second opinion that confirms that these plots are hard to interpret even for someone without my affliction. It is clear that considerable thought has gone into these plots, but I am not convinced that the cyclic RGB colourmap is an appropriate way to represent the data. It is not an ideal choice for two reasons. First, it is not perceptually uniform, meaning that the eye is drawn to specific points on the scale (in this case cyan and yellow), meaning that pixels (and any features) with exactly that specific combination of crater densities will be more (artificially) obvious. Second, the underlying data is not cyclic—a portion of the colourwheel is not relevant, because you will never have small and large craters, but no medium craters. Whatever background the density map is overlaid onto is also potentially unhelpful; it’s very difficult to tell what is background and what is crater density information. To address my concerns (and improve accessibility), I make the following suggestions.

a. Please consider an alternative way to plot the crater density data that uses a perceptually uniform colour scale. My tentative suggestion would be to try an approach that uses opacity (alpha) to depict total crater density and colour (on a perceptually uniform colour scale) to show some measure of the characteristic “size” of craters (the median crater size, perhaps?). Another suggestion would be to just plot the green channel. For Tooting, at least, I don’t see much (if anything) more in the full colour image than I do in the green.

b. Regardless of the final approach settled on, the method used to determine the colour of each pixel in the figures based on the raw crater counts should be explained in the methods section. For example, ED Fig. 5 suggests to me that the different colour channels are either on or off depending on whether a certain crater density is exceeded. Is that correct? Or is the value of each channel proportional to crater density as suggested in the text? Also, have the crater densities been normalised to account for the fact that craters in the middle size range (green) are more abundant (globally) than the larger (red) or smaller size range (blue)? Is there an upper and/or lower bound on the crater density for which any colour is plotted? I assume so as there are no black regions. Finally, I am surprised that one of the channels is craters smaller than 75-m given the poor recall and precision of the detections below 70 m. Although I agree that this does not appear to be critical for the purpose of identifying the secondary crater fields, I think it should be explicitly acknowledged that the crater densities in this channel are very inaccurate.

c. It would be very helpful to include a legend in the images to help show how to interpret the colors in terms of density and size. And I would also consider adding annotations/labels (and or text in the captions) to draw readers attention to the radial patterns identified.

d. It may also be helpful to reconsider the background on which the crater density map is overlain.

We really appreciate this constructive comment and the suggestions proposed to improve the visualization of this map. This is one of the central point of this work. We modified the rendering of our RGB map to simulate how readers suffering from deuteranopia see this figure. While major crater rays dominated by secondaries <75m (Tooting, Corinto, 12-000681, Resen...) are easily identifiable, the other craters are indeed not that obvious to pick.

Before detailing how we addressed this comment, I would like to draw your attention that an interface has been developed by our group to share the crater density map. I strongly encourage you to visit the following link to visualize the map on the globe at high resolution: <http://craters.computation.org.au/>. A high-resolution version of the map is also available at the following link and has also been added as a Supplementary Dataset : https://www.dropbox.com/s/birm7vfj5nofh52/HR_density.jpg?dl=0. You will have to sign up before accessing the data. The identification of secondary crater rays presented in this work has been performed through a careful survey, by going through this map, by zooming in and out. The static close-up images presented in the submitted version of the paper (ED. Fig. 8) is certainly not the most obvious way to identify most of the rays and their associated craters. In the revised manuscript, we included a Data availability section and mention the above link. We are still working on this interface, this includes the legend display and pixel labels depicting the local crater density. At the time of writing, we experience some issues regarding signing up of new users. This will be fixed as soon as possible. In case if you are not able to signup, below is a screenshot of what the interface looks like:

Regarding the map presented in the submitted version, we manipulated the data in multiple ways until finding the best combinations of crater size bin, colour stretch and resolution to enhance the secondary crater rays appearance. The size bins (25m-75m, 75m-150m, 150m-300m) and resolution (0.05 x 0.05 °/px) presented in the submitted version were found to be the best options for our purpose. The colour stretch was not presented; we clarified this point in the revised manuscript and acknowledged in the main text that the density of 25m-75m craters is not accurate, thus addressing the point (b): All the values of each band in the histogram beyond the 2nd standard deviation of the distribution are pushed to the end, becoming 0 or 255. The histogram is then redistributed to spread the remaining values from 0 to 255. This is done for each of the 3 RGB band (see updated ED Fig.5). The three bands are then merged together. In the resubmitted version, we addressed the points, (c) and (d) by modifying the background (MOLA shaded relief) and adding a legend allowing to interpret the crater density and size on each of these maps. A method section dedicated to the crater density map computation has been added in the revised manuscript.

Regarding the point (a), the middle of ED Fig.5 shows the crater density for the 75m-150m diameter range. I agree with the reviewer, this is sufficient to identify Tooting's rays. However, most of the other craters, in particular those smaller than 10km are not identifiable if the density of craters <75m is not considered. On the other hand, removing the band depicting the density of craters in the 150-300m diameter (for which, I agree, no major rays are identifiable) has the effect to (1) diminish the contrast between the over cratering areas and the cratering background and (2) remove the possibility to see the decreasing in small crater diameter (25m-300m) with the distance to a primary crater. This is observed for Tooting for instance. This observation is beyond the scope of the present study, let for future investigation and related to the 4th comment below. As you suggested, we have also tried to plot the median size of craters contained within each pixel. This computation is particularly intense if performed on a grid of 0.05°x0.05°. For instance, it took about 48 hours to compute on a descent laptop over the area surrounding Tooting crater: from -155°E to -145°E and from 18°N to 26°N. The results are shown below (left panel) and compared to the RGB crater density map (right panel):

Although we have to admit that this visualisation is interesting as it brings a new way to visualize our data and some patterns would certainly emerge from it, building and merging a global median crater size map with an alpha channel depicting the density would require the using of our HPC. For internal reasons, this is not possible at the moment. However, we will make it in the near future and will implement the result into the interface designed to present the CDA results mentioned above (<http://craters.computation.org.au/>).

Although our ability to update the colour schemes of the Figure 1 is limited, we refer now to the Extended Data Fig. 5 in the caption of Figure 1. The ED Fig. 5 has been updated using the three maps depicting the crater density for the three diameter range (available at the address mentioned above). This would allow to visualise all crater rays on at least on one of the maps. We also modified the Figure 1 to highlight the rays.

(3) Regarding the crater counts of the 19 candidate mars meteorite source craters. Although this may appear obvious to cratering experts, I think it would be helpful to be more explicit about which craters are used to date the crater ejecta blankets and the surrounding terrain, for the benefit of general readers. In particular, I presume that these are not the same craters as have been used to identify the secondary crater fields. If they are, some explanation of how self-secondary craters (i.e., secondary craters on top of the ejecta blanket) are dealt with might be important.

I agree, this point deserves to be clarified in both the main text and the Methods:

L.106: “We hypothesize that these 19 impact craters may represent the complete record of large and recent impact craters on the surface of Mars. To test this hypothesis, the ages of these 19 young impacts are determined using manual crater counts on their ejecta blanket (see Methods).”

L.120: “This approach assumes that crater counts on the area surrounding the crater source may provide the age of the volcanic episode responsible for the crystallization of the meteorite. Surrounding terrain model ages have been derived for all impact craters using the recently revised Mars crater catalogue for $D \geq 1 \text{ km}^{11-12}$, completed for higher precision by manual mapping of smaller craters if necessary (Methods).”

L.409: “We derived the formation model age of the 19 impact craters exhibiting rays of small secondaries ($< 300 \text{ m}$) by counting craters superposed on their well-defined ejecta blankets and excluded areas corresponding to the impact crater and its rims⁴³ using the CraterTools software⁴⁴. In order to determine the age of these young craters, we need to an accurate crater count superposed on the ejecta of these craters⁴⁵. Given the young age of these surfaces, a small number of craters are

expected, and the result is very sensitive to any mis-identification of craters (missing craters or false detections). Therefore, CDA results were manually checked and used as a first pass to pinpoint the impact craters superposed on each ejecta blanket and were completed by a manual count. The age of terrains surrounding each large crater has been derived using the manual crater database complete down to 1 km in diameter¹¹⁻¹² where secondary craters are marked and discarded from the counting.”

(4) The purpose of the modelling shown in ED Fig. 9 is unclear and could be better linked to the story. The models are perfectly valid, but the only thing they appear to inform is the total volume of displaced material in the Tooting impact. If this number is significant, then it would be good to explain why. One potentially interesting observation might be to compare this volume to the total volume of ejecta fragments required to produce the secondary craters.

Response: We clarified this point in the main text: L. 179: “Tooting crater ($D \approx 30$ km) is the second largest of the 19 rayed craters younger than 10 Myr old. Impact simulations made the formation of the Tooting crater (Methods and Supplementary Fig. 9) predict the excavation of a volume of $\sim 1.28 \times 10^{12}$ m³ (Supplementary Fig. 10), assuming excavation was made by a vertical impact direction. Similarly, the impact simulations were made for five other crater candidates. Supplementary Fig. 9 shows the moment when the transient crater was reached in each case. The transient crater cavity volume was used as an estimate the volume of excavated materials, summarised in Supplementary Fig. 10. Here we used the total excavated volume to quantify the increasing of the amount of ejected material with the crater diameter. This is relevant assuming a constant fraction collapses back into the crater and a fraction is ejected out of the crater, potentially forming secondary craters. Compared to the two other craters with young surface ages, ($\sim 3.97 \times 10^{11}$ m³ and $\sim 6.67 \times 10^{10}$ m³ for 09-000015 ($D \approx 20$ km) and Zunil ($D \approx 10$ km) respectively), Tooting has set in motion ~ 3.2 and ~ 19.3 times more material. This can be transposed to a relatively larger volume available for secondary cratering and rock ejection escaping Mars’ gravity compared to those related to 09-000015 and Zunil.”.

Regarding your suggestion about comparing rim volumes with that of the total fragments required to produce all of the secondaries detected by the CDA, although beyond the scope of this study, this idea and subsequent implications are currently under investigation by our team and would greatly benefit from inputs of experts in 3D impact modelling.

Congratulations to the authors for an excellent contribution that I hope to soon see in press.

Sincerely,

Gareth Collins

g.collins@imperial.ac.uk

REVIEWER COMMENTS

Reviewer #1 (Remarks to the Author):

The authors have successfully answered all of my comments. Congrats on a nice manuscript.

Reviewer #3 (Remarks to the Author):

The authors have addressed each one of my concerns in detail. In my opinion the manuscript is ready for publication if the following matter can be clarified:

“This labelled dataset was also augmented by applying a range of transformations (rotate, shear, scale, and translate) using YOLOv3(41).”

How many samples were produced/used in total?

Response: The repository where the samples were stored experienced an unexpected purge in April 2021. Unfortunately, I am afraid we are not able to retrieve this information anymore. Also, for internal reasons related to access to the supercomputer resources, we are not able to reproduce the augmentation to get this information at the time of writing. Nevertheless, the CNN was trained on 300 epochs over 75% of the marked tiles, which gives about 200,000 samples for the augmented dataset in total.

Were only the augmented training data lost? I understand that you may not have access to the resources you had during your research. As long as the original training dat set is available, the results can be reproduced and the mentioned data loss is not that significant. If the original dataset is lost, the results would not be reproducible, and that could be a problem. Could you please clarify this?

Here are a couple of typos I noticed:

“Identification of rayed craters is possible using images in the thermal infrared domain^{5,16}, but this approach is hampered by both image resolution (100 m/pixel), dust coverage (about half the Martian surface would remain not accessible by this technique) and physical properties of the impacted terrain.”

Could you refer to one or two of the physical properties that you mention? Also “both” is mostly used for 2 things, so here it would be better to erase it.

Response: The influence of the terrain’s physical properties on rays identification on thermal images is not well documented. We removed this part of the sentence.

Please replace “,” with “and” then.

“To visualize secondaries, and therefore recent primary of different sizes, the density of secondaries for three ranges of sizes ($150 < D < 300$ m, $75 < D < 150$ m and < 75 m) are represented as a map (Fig. 1) using respectively the red, blue and channels”

Please add: ...primaries...red, blue and green channels...

“To that end, we broke down all the processing steps of the previous version of the CDA17 to individual tasks, all with their own container that can be run individually and in parallel for different images. Each of which goes through all the individual steps producing crater locations in geographical coordinates of the original GeoTIFF image (using the spatial reference system specified therein).”

This part is confusing: Processing steps - individual tasks with their own container - each goes through all steps- So were the processing steps broken down to individual tasks at the end?

Response: I agree that this part was a confusing. We have clarified the presentation of these processing steps, L. 278: “To that end, we broke down all the processing steps (image reprojection and downsampling, tiling, scoring) of the previous version of the CDA17 to individual tasks, all with their own container. Thus, each of the task that can be run individually and in parallel for different images.”.

You mean “each of the tasks can ...

I appreciate the high quality of your work and the thorough and detailed responses during the review process.

Sincerely,

Lida Fanara

lida.fanara@dlr.de

German Aerospace Center (DLR)

Institute of Planetary Research

Reviewer #4 (Remarks to the Author):

Dear Authors,

I am grateful to you for making such a concerted and diligent effort to address my comments from the first version of the manuscript. The manuscript is much improved. In particular, your efforts to improve the clarity of Figure 1 and the relevant supporting figures are both welcome and effective. Together with the supplementary data, I am satisfied that the MS now provides compelling justification for the conclusions. I am delighted to recommend publication of the manuscript in its present form (pending two very minor further suggestions).

Minor suggestions:

One suggestion from my previous review that has not been implemented is the grammar in the second sentence of the abstract. It should either be “Their parental liquid, ... , where it eventually

crystallised” or “Their parental liquids, ... , where they eventually crystallised.” If the singular is chosen, the following sentence would need to be made consistent.

Second, since reading your manuscript I came across a paper by Mouginis-Mark et al., 1992, which I think should be briefly discussed and referenced in the MS. While it does not mention Tooting by name (I think the name was acquired later), it does discuss the strengths and weaknesses of “Crater 2” (Tooting crater) as a candidate for the source of the shergottite meteorites.

Mouginis-Mark, P.J., McCoy, T.J., Taylor, G.J., Keil, K., 1992. Martian parent craters for the SNC meteorites. *Journal of Geophysical Research: Planets* 97, 10213–10225.
<https://doi.org/10.1029/92JE00612>

With warm regards,

Gareth Collins

Reviewer #2 (Remarks to the Author):

Review of revised manuscript: “The Tharsis mantle source of depleted Shergottites revealed by 90 million impact craters” **for Nature Communications, by Lagain et al.**

This manuscript studies the crater source and source of the depleted shergottite martian meteorites using the novel Crater Detection Algorithm (CDA) and finds that the Tooting crater was the likely source crater of these rocks. This study is important as we currently do not know where the martian meteorite originate from at the martian surface. In addition, depleted shergottites are some of the most common martian samples. The CDA can also be used for other studies focusing on craters of different sizes. Most of my comments were addressed by the authors. This revised manuscript is well-written and the implications are well-justified. I recommend publication after very minor revisions:

Make sure to check that “shergottites” is not capitalized

Include a space between the abbreviation and number of meteorites (for example NWA 7034 and not NWA7034)

L83-84. I am still not sure if I understand the statement “Thus, the crater source of depleted shergottites launched 1.1 Myr ago should be associated with abundant secondaries in the ~10 - 300 m-size range.” According the previous sentences, are the 10–300 m size is according to the age of the crater and not the size of the primary crater? Please explain (for the non-crater experts, like me)

For the supplementary figure 2, I did not mean to redo it but to check the resolution. The resolution is poor on the version I currently have and thus it is hard to distinguish the different areas.

Arya Udry

Reviewer #1 (Remarks to the Author):

Dear Dr. Filiberto,

Thank you very much for your feedback. We appreciated your thorough reviews and constructive comments that greatly improved both form and substance of our manuscript.

The authors have successfully answered all of my comments. Congrats on a nice manuscript.

Reviewer #2 (Remarks to the Author):

Dear Dr Udry,

Thank you very much for your feedback. We appreciated your thorough reviews and constructive comments that greatly improved both form and substance of our manuscript.

This manuscript studies the crater source and source of the depleted shergottite martian meteorites using the novel Crater Detection Algorithm (CDA) and finds that the Tooting crater was the likely source crater of these rocks. This study is important as we currently do not know where the martian meteorite originate from at the martian surface. In addition, depleted shergottites are some of the most common martian samples. The CDA can also be used for other studies focusing on craters of different sizes. Most of my comments were addressed by the authors. This revised manuscript is well-written and the implications are well-justified. I recommend publication after very minor revisions:

Make sure to check that “shergottites” is not capitalized

Response: Agree and corrected.

Include a space between the abbreviation and number of meteorites (for example NWA 7034 and not NWA7034)

Response: Agree and corrected.

L83-84. I am still not sure if I understand the statement “Thus, the crater source of depleted shergottites launched 1.1 Myr ago should be associated with abundant secondaries in the ~10 – 300 m-size range.” According the previous sentences, are the 10–300 m size is according to the age of the crater and not the size of the primary crater? Please explain (for the non-crater experts, like me)

Response: This paragraph (L.65-88) explains in details why the presence of such small secondaries attests of the freshness of their associated large primary crater.

- The erosion rate on Mars mentioned L.76 leads to the rapid obliteration of small impact structures on the surface.
- As mentioned L. 73, an impact forming a 30 km crater on Mars produces secondaries smaller than 1 km in diameter.

The age and size of the primary craters warrant identification and analysis of the distribution of the small secondary craters. The size range of 10-300 m is an acceptable estimate for diagnosing a recent primary impact crater from which the small secondary craters originated. This sentence now reads: “Therefore, the occurrence of radial patterns of small secondaries associated with a primary crater is a diagnostic feature of a recent impact^{6,15-16}. Thus, the crater source of depleted shergottites launched 1.1 Myr ago should be associated with abundant small secondaries, in the ~10 - 300 m-size range.”

For the supplementary figure 2, I did not mean to redo it but to check the resolution. The resolution is poor on the version I currently have and thus it is hard to distinguish the different areas.

Response: The supplementary figure 2 illustrates that the crater diameter estimated by the CDA is independent from the terrain type and that the diameter of craters > 70 m is estimated within 25% of errors. It is not necessary to distinguish each data point here. The resolution seems to be fine, I attached below the current version of the figure.

Arya Udry

Reviewer #3 (Remarks to the Author):

Dear Mrs Fanara,

Thank you very much for your feedback. We appreciated your thorough reviews and constructive comments that greatly improved both form and substance of our manuscript.

The authors have addressed each one of my concerns in detail. In my opinion the manuscript is ready for publication if the following matter can be clarified:

“This labelled dataset was also augmented by applying a range of transformations (rotate, shear, scale, and translate) using YOLOv3(41).”

How many samples were produced/used in total?

Response: The repository where the samples were stored experienced an unexpected purge in April 2021. Unfortunately, I am afraid we are not able to retrieve this information anymore. Also, for internal reasons related to access to the supercomputer resources, we are not able to reproduce the

augmentation to get this information at the time of writing. Nevertheless, the CNN was trained on 300 epochs over 75% of the marked tiles, which gives about 200,000 samples for the augmented dataset in total.

Were only the augmented training data lost? I understand that you may not have access to the resources you had during your research. As long as the original training data set is available, the results can be reproduced and the mentioned data loss is not that significant. If the original dataset is lost, the results would not be reproducible, and that could be a problem. Could you please clarify this?

Response: The original training dataset (before augmentation) is available at the following address: <https://doi.org/10.5281/zenodo.5514314>. It contains the tiled images and labels required to retrain the CNN as well as the name of the tiles used for both training and validation. We also double-checked the number of craters manually mapped to build the training dataset: 2142. The previous version of the manuscript stated 2200 craters. We updated this value L. 278.

Here are a couple of typos I noticed:

“Identification of rayed craters is possible using images in the thermal infrared domain^{5,16}, but this approach is hampered by both image resolution (100 m/pixel), dust coverage (about half the Martian surface would remain not accessible by this technique) and physical properties of the impacted terrain.”

Could you refer to one or two of the physical properties that you mention? Also “both” is mostly used for 2 things, so here it would be better to erase it.

Response: The influence of the terrain’s physical properties on rays identification on thermal images is not well documented. We removed this part of the sentence.

Please replace “,” with “and” then.

Response: Agree and corrected: “,” has been replaced by “and”.

“To visualize secondaries, and therefore recent primary of different sizes, the density of secondaries for three ranges of sizes ($150 < D < 300$ m, $75 < D < 150$ m and < 75 m) are represented as a map (Fig. 1) using respectively the red, blue and channels”
Please add: ...primaries...red, blue and green channels...

Response: Agree and corrected: we added “green” L.98.

“To that end, we broke down all the processing steps of the previous version of the CDA17 to individual tasks, all with their own container that can be run individually and in parallel for different images. Each of which goes through all the individual steps producing crater locations in geographical coordinates of the original GeoTIFF image (using the spatial reference system specified therein).”

This part is confusing: Processing steps - individual tasks with their own container - each goes through all steps- So were the processing steps broken down to individual tasks at the end?

Response: I agree that this part was a confusing. We have clarified the presentation of these processing steps, L. 278: “To that end, we broke down all the processing steps (image reprojection and downsampling, tiling, scoring) of the previous version of the CDA17 to individual tasks, all with their own container. Thus, each of the task that can be run individually and in parallel for different images.”.

You mean “each of the tasks can ...

Response: Agree and corrected: we replaced “task” by “tasks” L.290.

I appreciate the high quality of your work and the thorough and detailed responses during the review process.

Sincerely,
Lida Fanara

lida.fanara@dlr.de

German Aerospace Center (DLR)
Institute of Planetary Research

Reviewer #4 (Remarks to the Author):

Dear Dr. Collins,

Thank you very much for your feedback. We appreciated your thorough reviews and constructive comments that greatly improved both form and substance of our manuscript.

Dear Authors,

I am grateful to you for making such a concerted and diligent effort to address my comments from the first version of the manuscript. The manuscript is much improved. In particular, your efforts to improve the clarity of Figure 1 and the relevant supporting figures are both welcome and effective. Together with the supplementary data, I am satisfied that the MS now provides compelling justification for the conclusions. I am delighted to recommend publication of the manuscript in its present form (pending two very minor further suggestions).

Minor suggestions:

One suggestion from my previous review that has not been implemented is the grammar in the second sentence of the abstract. It should either be “Their parental liquid, ... , where it eventually crystallised” or “Their parental liquids, ... , where they eventually crystallised.” If the singular is chosen, the following sentence would need to be made consistent.

Response: Agree and corrected: we replaced “liquid” by “liquids” L.23.

Second, since reading your manuscript I came across a paper by Mouginis-Mark et al., 1992, which I think should be briefly discussed and referenced in the MS. While it does not mention Tooting by name (I think the name was acquired later), it does discuss the strengths and weaknesses of “Crater 2” (Tooting crater) as a candidate for the source of the shergottite meteorites.

Mouginis-Mark, P.J., McCoy, T.J., Taylor, G.J., Keil, K., 1992. Martian parent craters for the SNC meteorites. *Journal of Geophysical Research: Planets* 97, 10213–10225. <https://doi.org/10.1029/92JE00612>

Response: Indeed, Tooting crater was named in 2006 after a long debate between Peter Mouginis-Mark and the IAU. We are aware about his 1992 paper and we thank the reviewer for pointing this out. In our manuscript we refer to a more recent paper published by this author in 2012 (REF 7, see below), stating in our introduction L.40: “The ejection sites are still unknown, despite several previous propositions⁵⁻⁷,”. We also refer to his 2007 paper and his 2015 geological map (REFs 24 and 27 respectively). Together those three papers update and detail his 1992 investigations on Tooting crater. We do not believe it is required to cite this old paper as further iteration exist and were published by the same author.

With warm regards,
Gareth Collins

REVIEWERS' COMMENTS

Reviewer #3 (Remarks to the Author):

Thank you for clarifying that the training data set was not lost and for making it available!

Also thank you again for the nice work and careful revision!

Sincerely,

Lida Fanara